# Discriminator-Guided Embodied Planning for LLM Agent

**Haofu Qian**[1], **Chenjia Bai**[2,4,*], **Jiatao Zhang**[1,3], **Fei Wu**[1], **Wei Song**[1,3,*], **Xuelong Li**[2]

[1]Zhejiang University [2]Institute of Artificial Intelligence (TeleAI), China Telecom,
[3]Yuhang Humanoid Robot Industry Innovation Center, Hangzhou, China,
[4]Shenzhen Research Institute of Northwestern Polytechnical University
qianhf@zju.edu.cn, baicj@chinatelecom.cn

## Abstract

Large Language Models (LLMs) have showcased remarkable reasoning capabilities in various domains, yet face challenges in complex embodied tasks due to coherent long-term policy, context-sensitive environmental understanding. Previous work performed LLM refinement relying on outcome-supervised feedback, which can be costly and ineffective. In this work, we introduce a novel framework, **D**iscriminator-**G**uided **A**ction OP**timization (DGAP) for facilitating optimization of LLM action plans via step-wise signals. Specifically, we employ a limited set of demonstrations to enable the discriminator in learning a score function, which assesses the alignment between LLM-generated action and the underlying optimal one at every step. Based on the discriminator, LLM is prompted to generate actions to maximize the score utilizing historical action-score pairs trajectory as guidance. Under mild conditions, DGAP resembles the critic-regularized optimization and is demonstrated to achieve a stronger policy than the LLM planner. In experiments across different LLMs (GPT-4, Llama3-70B) in ScienceWorld and VirtualHome, our method obtains superior performance and better efficiency than previous methods.

## 1 Introduction

The effectiveness of large language models (LLMs) in task planning hinges on their ability to generate coherent, executable plans in dynamic, open-ended environments (Song et al., 2023; Suzgun et al., 2022), However, embodied scenarios pose greater challenges, requiring long-term planning in intricate contexts with high-dimensional state spaces and tangible interactions. Specifically, it involves coordinating long action sequences while managing intricate, dynamic environments, initial inaccuracies compounding over steps, causing significant deviations from the plan and risking mission failure (Ross et al., 2011; Luo et al., 2024).

Facing these difficulties, in-context learning methods (Dong et al., 2022; Abernethy et al., 2023; Akyürek et al., 2023), tree-of-thought(ToT) methods (Yao et al., 2023a; Feng et al., 2023; Zhou et al., 2023a) and demonstration-based methods (Lin et al., 2023; Rita et al., 2024) have accomplished partial progress. However, a key issue is that they generally receive feedback signals at the trajectory level, which is non-proactive and limits their effectiveness and generalization in dynamic embodied scenarios (Chen et al., 2024b; Liu et al., 2024; Shi et al., 2024). In particular, in-context learning methods introduce closed-loop feedback(Song et al., 2023; Wu et al., 2023) for failed results at completion via inner monologue (Madaan et al., 2023; Huang et al., 2022a; Yao et al., 2023b) or physical feedback (Shinn et al., 2023; Mandi et al., 2023). ToT methods (Yao et al., 2023a; Feng et al., 2023; Zhou et al., 2023a) generate multiple trajectories to represent several reasoning pathways, and perform trajectory-level switching optimization, which incurs high exploration costs (Zhang et al., 2024a). Demonstration-based approaches require extensive, high-quality trajectories in diverse scenarios to obtain a generalized policy (Lin et al., 2023; Rita et al., 2024) (see Fig. 1).

To address these limitations, we consider an alternative way for grounding and generalization of embodied planning by leveraging the knowledge from limited demonstrations to construct step-level guidance, which is subsequently integrated into in-context learning to boost planning. A key

---

*Corresponding Author

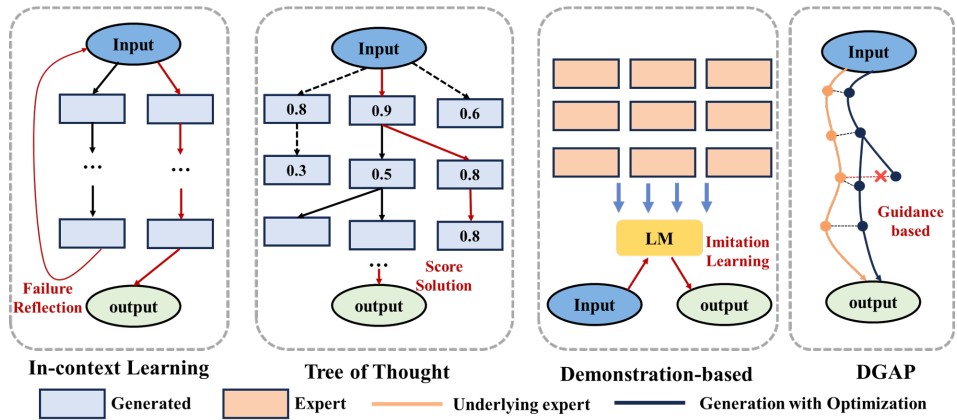

Figure 1: Comparing methods of LLMs to conduct embodied planning. DGAP leverages step-wise alignment signals to guide LLMs in the planning process.

challenge is applying the limited information from few-shot demonstrations in diverse scenarios. As described before, embodied agents that directly imitate these demonstrations often accumulate errors due to their restricted scope (Ross et al., 2011; Luo et al., 2024). In contrast, humans possess distinct capabilities for internal generalization and alignment (Barsalou, 2008; Rizzolatti & Craighero, 2004), allowing them to comprehend quickly task objectives and environment dynamics from few-shot demonstrations (Schaal, 1999; Malaviya et al., 2022). Motivated by this, we allow the agent to acquire a small number of demonstrations and further leverage them to provide step-wise signals to embodied planning. Specifically, (i) the demonstrations includes task objective and dynamics constraints that are lacking in LLMs, while it struggles to handle Out-Of-Distribution (OOD) situations across diverse scenarios; and in contrast, (ii) LLMs that contain a vast amount of commonsense knowledge enabling long-term reasoning on a broad set of situations without suffering from OOD problems, while LLMs lack domain knowledge and expert guidance for grounding. As a result, our work aims to combine the benefits of LLMs with the precise insights from demonstrations to provide few-shot generalization and task-specific grounding simultaneously.

In this paper, we propose a novel framework named Discriminator-Guided Action Optimization (DGAP) to perform efficient grounding and generalization for the LLM planner at every step. Instead of imitating expert actions, we learn a discriminator on a small number of demonstrations to form a score function within the framework of sentence transformers, which measures the alignment between LLM actions and underlying expert actions at the step level. By using meta-prompts that contain previously generated actions with score guidance Yao et al. (2023a), the LLM planner performs as a closed-loop optimizer for new actions with high scores through its understanding of existing discriminative action-score pairs. This approach effectively treats the LLM as an optimizer (Yang et al., 2024), which inherently involves an optimizing process, bypassing the need for physical or outcome feedback required by previous methods for refinement. Under mild conditions, we show that DGAP resembles the critic-regularized optimization in RL and is provable to achieve a stronger policy than the LLM planner.

Our contributions are summarized as follows. (i) We propose a novel framework that combines the long-term reasoning of LLMs and task-specific grounding under guidance from a small number of demonstrations. (ii) We propose a simple discriminator learned with a small number of demonstrations to serve as a step-level score function, which helps the LLM planner generate high-score actions via implicit optimization. (iii) We build theoretical connections between DGAP and critic-regularized optimization in RL, which shows our method obtains a stronger policy than the LLM planner under mild conditions. (iv) We conduct extensive experiments in challenging ScienceWorld (Wang et al., 2022) and VirtualHome (Puig et al., 2018) benchmarks combined with different LLM planners (GPT-4 (OpenAI, 2023), Llama3-70B (Meta, 2024)), and the result shows DGAP obtains superior performance and better efficiency than previous methods.

## 2 PRELIMINARIES

The embodied tasks can be formalized as a Markov Decision Process (MDP) defined by a tuple $\mathcal{M} = (\mathcal{L}, \mathcal{S}, \mathcal{O}, E, \mathcal{A}, \mathcal{P}, r, T)$. In the MDP, $l \sim \mathcal{L}$ is the language description of a task specifying a high-level goal. For example, in ScienceWorld, a task description can be "*your task is to boil lead*".

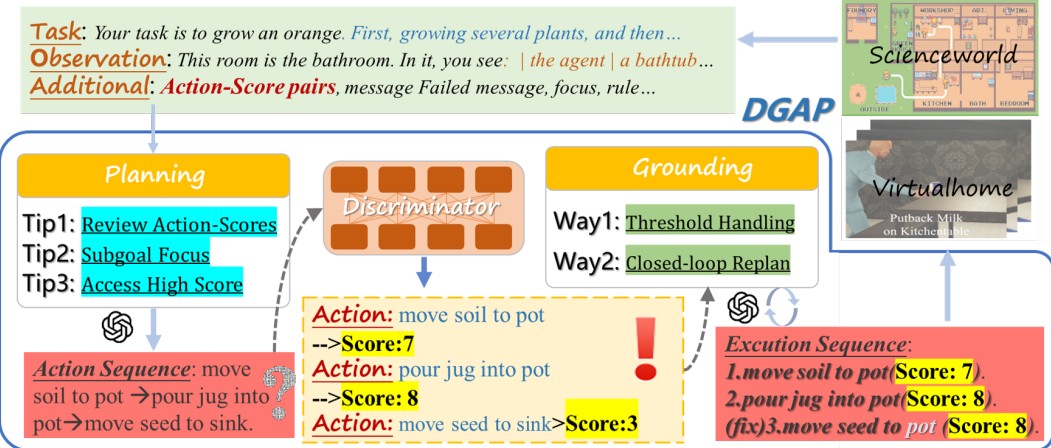

Figure 2: Framework of DGAP: **(i)** Acquiring domain-specific knowledge through discriminator's regressive training. **(ii)** Optimizing action generation via historical action-score pairs.

At time step $t$, the state $s_t \sim \mathcal{S}$ contains the description of environment information including the agent observations $o_t \sim \mathcal{O}$ (i.e., environment feedback of previous action or information queried) and environment states $e_t \sim E$ (i.e., details about all visible objects). The action $a_t \sim \pi(a|s_t)$ is generated by following a valid policy $\pi : \mathcal{S} \to \Delta_{\mathcal{A}}$. For example, an LLM planner $\pi^{\text{llm}}(a_t|s_t)$ is a valid policy that generates an action $a_t$ based on the state $s_t$. In most embodied planning tasks, a valid action should follow some supported action templates such as *"use X on Y, examine Y"* For example, a valid action $a_t$ in this task can be *"use thermometer on liquid tin,"* and the initial action $a_0$ is always *"look around"* for showing initial environment information. Each step involves the agent executing the action to interact with the environment, obtaining the next state $s_{t+1}$ and some reward $r_t$. For an RL-based agent, the rewards will be used for policy learning via policy gradient algorithms. As for an LLM policy, the policy generates actions based on commonsense knowledge, the provided prompts, and (optionally) the closed-loop feedback.

In DGAP, we additionally acquire a small number of demonstrations, which include the optimal action sequences generated by an oracle planner $\pi^{\text{oracle}}$. We denote the expert dataset as $\mathcal{B} = \{l, (a_0^i, a_1^i, \ldots, a_T^i)\}_{i=1}^M$, where $M$ denotes the number of episodes. An imitation policy $\pi^{\text{bc}}$ is learned via Behavior Cloning (BC) in the expert data, by maximizing the log-likelihood of expert actions as $\pi^{\text{bc}} = \max_\pi \mathbb{E}_{\mathcal{B}}[\pi(a_t^i|l, h_t)]$, where $h_t = a_{t-1}^i, ..., a_{t-n}^i$ contains previous actions up to 10. $\pi^{\text{bc}}$ can perform poorly in real-world interactions due to limited state coverage of demonstrations.

## 3 METHOD

In this section, we present the details of our proposed framework, DGAP, as depicted in Fig. 2. This framework capitalizes on the domain knowledge embedded in both expert and handcrafted datasets to assign scores to responses from the LLM. These scores then strategically guide LLMs in planning towards specified objectives. Specifically, it consists of three parts: *(i)* Acquiring domain-specific knowledge through discriminator's regressive training from augmented expert data with score labels in §3.1. *(ii)* Utilizing the pre-trained discriminator to optimize action generation via historical action-score pairs and ground actions when facing anomalous scores in §3.2. *(iii)* Qualitative analysis of DGAP and critic-regularized optimization.

### 3.1 THE DISCRIMINATOR AS SCORER

As previously mentioned, reflection-based methods (Madaan et al., 2023; Shinn et al., 2023) and search-based methods (Yao et al., 2023a; Chen et al., 2024a; Zhou et al., 2023a) supervised by outcome present inefficient to some extent. Demonstration-based methods require large datasets for their scalability in embodied tasks (Rita et al., 2024; Sun & van der Schaar, 2024). Recognizing the potential complementarity of them, we investigate a solution that combines their strengths by employing a discriminator. Specifically, the discriminator integrates external knowledge from demon-

strations, subsequently provides a *score* measuring the alignment between the LLM's response and the underlying expert choice within the current context. The process is defined as:

$$\mathcal{D}_\phi : (l, h_t, a_{\pi^{\text{llm}}(l,s_t)}) \rightarrow Q, \tag{1}$$

where $D_\phi$ denotes the offline discriminator with parameters $\phi$, $l, s_t$ refers to task objective and environmental information as stated in Sec. 2, $h_t$ is a summary of the past ten actions, $a_{\pi^{\text{llm}}(l,s_t)}$ represents the action generated by LLM based on task goal and state at timestep $t$. $Q$ is the score (between 0 and 10), where a higher score indicates greater alignment with the expert policy. A detailed description of the discriminator is in Appendix C.

Previous research has highlighted the generalization capacity and adaptability of demonstrations in planning for various embodied tasks (Mu et al., 2023; Lin et al., 2023). Unlike existing methods that directly utilize fixed demonstrations to transfer to new scenarios, our approach converts the demonstrations' information into numeric values, enabling a measurable step-level feedback. Additionally, We enhance these values through augmentation on limited demonstration samples to improve their generalization across embodied scenarios. This numeric representation is used to simplify integration with task planning, offering a more efficient, scalable solution with dense feedback.

**Data Overview**  The discriminator is designed to numerically differentiate actions. Intuitively, embedding regression serves as an effective method for this purpose. A predominant factor contributing to this is the high similarity between expert action embeddings and those of generated actions, which makes it difficult to distinguish and be represented as a score with generalization. To tackle this issue, previous studies propose data distribution adjustments such as an unbalanced combination of expert data, sub-optimal data (Xu et al., 2022), and data augmentation to provide generization (Jha et al., 2020; Zhang et al., 2024b) utilizing a customized fine-tuned LM (Tan et al., 2024). In light of them, we adopt a data collection strategy that involves a tailored modification to expert data, combined with comparable random data and a substantial volume of generated data via fine-tuned LM, as illustrated in Fig. 3. And the specific implementing details are stated in Appendix. C.

- **Expert Data (score 10)**: This parts are composed of oracle trajectories from ScienceWorld (Lin et al., 2023) and VirtualHome (Puig et al., 2018) official datasets, from which we exact $l$ and $h_t$ as instructions. The correlated instruction-action pairs adhere to policy $\pi^{\text{oracle}}$ and are assigned a score of 10. Together, they constitute dataset $\mathcal{B}_e$.

- **Random Data (score 0)**: We collect a dataset with negative pairs where the instruction is paired with a ground truth action from demonstrations that is the least semantically related, where both the predicate and the object of the action are changed and collectively form the dataset $\mathcal{B}_r$.

- **Offline Data (score within [0, 10])**: Accomplishing regression tasks using highly polarized samples presents considerable difficulty. To enrich the diversity of both instructions and actions, we fine-tune a language model (LM) via imitation learning (IL) to provide domain knowledge, using the instruction-action pairs in expert data, as detailed in Appendix B. Subsequently, we utilize a fine-tuned model to generate action candidates $a_{\pi^{\text{bc}}}$ through beam search across a range of instructions, thereby forming entirely new pairs. Here $a_{\pi^{\text{bc}}}$ denotes the action generated by the fine-tuned LM. We employ a pre-trained sentence embedding model to extract candidates' features and further evaluate semantic similarity with ground-truth actions $a_{\pi^{\text{oracle}}}$. This process is described as $Sim(a_{\hat{\pi}^{\text{bc}}}, a_{\pi^{\text{oracle}}})$. In unseen scenarios without ground-truth, we mildly treat the first candidate generated by the LM as 10 for the distribution of fine-tuned small models aligns more closely with the training data domain (Panigrahi et al., 2023). The scores for the subsequent candidates are determined by multiplying their cosine similarity to the first candidate by 10. These data hereby make up dataset $\mathcal{B}_a$ and constitute a significant proportion of the overall dataset $\mathcal{B}_o$.

The reason we transfer the similarity naturally ranges between $[0, 1]$ to integrals between $[0, 10]$ is that LLMs more effectively process integers than decimals (Gruver et al., 2024). Thus, during the discriminator's training, we scaled the scores to span in $[0, 10]$ and subsequently rounded them to integers for the subsequent stage of LLM reasoning to better optimize LLMs' interpretative effectiveness.

**Model Overview**  Previous research has demonstrated the significant capability of sentence transformers in performing text classification (Cohan et al., 2019). Similarly, We also formulate our discriminator's training as a regression task and employ another sentence transformer-based network

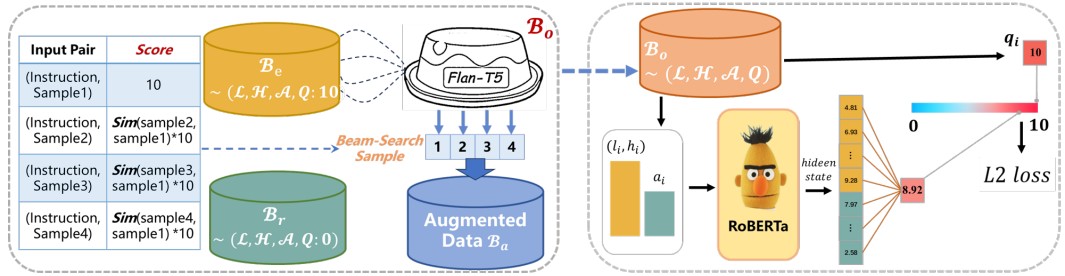

Figure 3: Illustration of dataset construction and discriminator training

as a backbone, utilizing the data in $\mathcal{B}_o$, which aims to establish the mapping from instruction-action pairs to scores. Specifically, we employ the RoBERTa (Liu et al., 2019) model architecture complemented by a linear head to get a precise score, thereby ensuring a robust integration of paired data into our evaluative framework. We explicitly apply L2 loss in training as shown in equation 2. Details are in Appendix C.

$$\mathcal{L} = \arg\min_{\phi} \left( \mathbb{E}_{(\mathcal{L}, \mathcal{H}, \mathcal{A}) \sim \mathcal{B}_o, \mathcal{A} \sim \pi^{\mathrm{bc}}} \left[ \sum_{i=1}^{n} \| \mathcal{D}_\phi(l_i, h_i, a_i) - \mathcal{Q}_i \|_2 \right] \right) \qquad (2)$$

The introduction of discriminator offers intuitive and readily obtainable feedback which supplants accessing feedback through exploration in the environment, and enhances the planning process by directly concentrating on the action-score information.

## 3.2 LLM OPTIMIZATION WITH DISCRIMINATOR

In this section, we outline how to combine the pre-trained discriminator with in-context learning to boost planning: *(i)* Prompts with scores to equip the LLM with the foresight to discern whether each action contributes to the successful completion of the task. *(ii)* Given the LLM's inherent role as a sampler (Zhao et al., 2023a; Hopkins et al., 2023), a closed-loop feedback is introduced to ensure the optimization process. When the action score at any step falls below a predefined threshold, the LLM planner is required to adjust its policy in response to the suboptimal performance observed in the prior iteration.

**Prompt with scores** Unlike the prevalent use of Outcome-Supervised Prompts, which assess only the final outcome of solutions (Hu et al., 2023; Zhou et al., 2023a; Shinn et al., 2023; Yao et al., 2023b; Wang et al., 2023c), our approach implements a Process-Supervised Prompt (PSP) via scores obtained at each step.

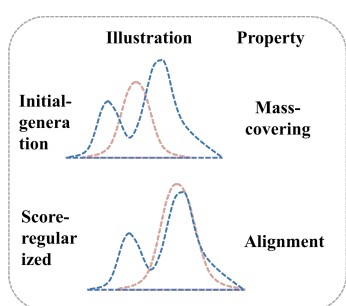

Figure 4: Regularizing the LLM policy distribution through score-based replanning

Through this way, LLMs are encouraged to model and optimize task completion by evaluating intermediate steps, enhancing its decision-making in complex, long-term tasks (Hao et al., 2024; Xiong et al., 2024). Specifically, LLMs are allowed to see the entire trajectory of executed action-score pairs, dynamically adjusting its next action to maximize future rewards. By focusing the planning on numerical feedback, the approach is similar to reinforcement learning (RL), where agents refine their policies through incremental action-based rewards, as to identify key decision points, evaluate potential future paths, and rebalance its plan according to the stepwise feedback. There are more details in Corollary 3.2. The structured feedback mechanism ensures that each action aligns actions more accurately with long-term objectives while reducing the risk of cumulative errors (Sun & van der Schaar, 2024; Rita et al., 2024), a challenge aforementioned in emboied planning.

Building on this theoretical foundation, we integrate step-based scores into the prompt structure. Several studies have demonstrated that LLMs' numerical sensitivity and utilization capabilities can be unveiled through appropriate prompting (Yang et al., 2024; Liu et al., 2023). Drawing inspiration from these findings, we have integrated the following context into our prompts to heighten

the model's focus and faithfulness on scores and further leverage LLMs' capacity to generate and optimize existing solutions. For a detailed prompt template, please refer to Appendix D.3.

> *Previous **actions, scores** and observations are as follows...*
> *You are required to **maximize high-score actions cumulatively** while adhere to the task*
> *Identify the **intrinsic relationship** between the action-score pairs*

**Refinement for the failed plan**   As an inherent role as a sampler (Zhao et al., 2023a; Hopkins et al., 2023), the LLM is required to function as an optimizer. Considering the diffculty of generating action by score maximizing implicitly and the hindrance caused by high variance in action-score pairs to the LLM, a closed-loop refinement process is introduced. If the score falls below the threshold $\tau$, the LLM is asked to replan based on the unsatisfactory action-score pair until $D_\phi(l_t, h_t, a_t \sim \pi_{\text{llm}}) > \tau$, as illustrated in Fig. 7. This ensures improvements over $\pi_{\text{llm}}$ by score, with minimal changes to its policy through the closed-loop replanning process. It also decreases the interaction rounds of agents since the action $a_t$ has been evaluated and refined via discriminator feedback before execution. Specifically, we adopt a threshold of 5 for ScienceWorld and 6 for VirtualHome, based on their respective training data distributions.

### 3.3   QUALITATIVE ANALYSIS

In the following, we give qualitative analysis to connect the proposed framework and the critic-regularized optimization problem. Since the score function $\mathcal{D}_\phi(s, a)$ measures the similarity between LLM actions and expert actions, it forms an implicit *reward function* for the LLM agent as $r_\phi = \mathcal{D}_\phi(s, a)$. In DGAP, since the planner is prompted to generate an action that maximizes the score function, our method implicitly maximizes the reward considered in the RL framework. Meanwhile, since the output of DGAP largely relies on the commonsense knowledge of the initial LLM planner (i.e., $\pi^{\text{llm}}$), the resulting policy still lies closely to the $\pi^{\text{llm}}$. As a result, our method can be formalized as a constrained optimization problem that aims to learn an improved policy $\pi^{\text{dgap}}$ over the initial $\pi^{\text{llm}}$ by maximizing rewards, as

$$\pi^{\text{dgap}} = \arg\max_{\pi_\theta} \mathbb{E}_{s_t \sim d_{\text{llm}}^\pi, a_t \sim \pi^{\text{llm}}} \Big[ \sum_t r_\phi(s_t, a_t) \Big] - \beta D_{\text{KL}}\big[\pi_\theta(a_t|s_t) \| \pi^{\text{llm}}(a_t|s_t)\big], \quad (3)$$

where the states are sampled from a state distribution $d_{\text{llm}}^\pi(s)$ induced by the LLM policy, the actions are sampled by following the LLM policy, and $\beta$ is a balance factor. The cumulative return is defined as $R_\phi(s_t, a_t) = \sum_{i=t}^{T-1} r_\phi(s_i, a_i)$ without a discount factor. Then the optimization objective becomes

$$\mathcal{L}^{\text{dgap}} = \mathbb{E}_{s \sim d_{\text{llm}}^\pi, a \sim \pi^{\text{llm}}}\big[ R_\phi(s, a) \big] - \beta D_{\text{KL}}\big[\pi_\theta(a|s) \| \pi^{\text{llm}}(a|s)\big]. \quad (4)$$

In DGAP, since $R_\phi(s, a)$ comes from a learned discriminator, it can be considered as the *critic* in an RL framework. We remark that such an objective is slightly different from DGAP where the LLM planner is prompted to generate actions that maximize the single-step return rather than the cumulative return since an episode-level discriminator can be more difficult to train. Nevertheless, for embodied planning tasks in ScienceWorld and VirtualHome, a successful multi-step plan requires single-step optimality in each planning step. The objective in equation 4 resembles a critic-regularized RL objective (Peng et al., 2019). Then the following Lemma gives the solution for $\pi_\theta$ that maximizes this objective.

**Lemma 3.1.** *The optimal policy that solves the constrained optimization problem in $\mathcal{L}^{\text{dgap}}$ is*

$$\pi_\theta^\star(a|s) = \pi^{\text{llm}}(a|s) \exp\big(R_\phi(s, a)/\beta\big)/Z(s), \quad (5)$$

*where $Z(s)$ is a normalized factor to make $\pi_\theta^\star(a|s)$ a valid policy, i.e., $\int_a \pi_\theta^\star(a|s)da = 1$.*

The proof is given in Appendix A. Then we have a direct corollary, which is as follows:

**Corollary 3.2.** *The updated policy $\pi_\theta^\star(a|s)$ improves over $\pi^{\text{llm}}(a|s)$ as $Q^{\pi_\theta^\star}(s, a) \geq Q^{\pi^{\text{llm}}}(s, a)$.*

The proof is given in Appendix A. In Corollary 3.2, the $Q$-function is defined as the expected return as $Q^\pi(s, a) = \mathbb{E}_\pi[R_\phi(s, a)]$. As a result, the policy $\pi_\theta^\star(a|s)$ is provable to obtain a higher expected

return than $\pi^{\mathrm{llm}}$ in the data coverage of the LLM policy. In practice, $Z(s)$ is a partition function that can be hard to estimate, so we have

$$\pi^{\mathrm{dgap}} \propto \pi^{\mathrm{llm}}(a|s) \exp\left(R_\phi(s, a)/\beta\right). \tag{6}$$

Thus, the resulting policy $\pi^{\mathrm{dgap}}$ combines the benefits of the LLM planner and score function. Specifically, LLM is a basic policy that gives candidates actions with high probability, and then the score function forms a filter to choose actions from the candidates with the highest scores. Such a process is simplified in DGAP by using LLM as an optimizer, finally obtaining better policies in expected returns. The parameter $\beta$ is a tuning factor that balances the effect of the LLM planner and score functions. In a special case, when $\beta \to \infty$, we have $\pi_\theta^\star \approx \pi^{\mathrm{llm}}$.

**Connection to RLHF.** Our method is also closely related to Reinforcement Learning from Human Feedback (RLHF) (Ouyang et al., 2022; Bai et al., 2022; Touvron et al., 2023), which follows a similar optimization objective as in equation 3 that optimizes some reward function with constraints to the supervised fine-tuning (SFT) model. However, there exist several major differences. (i) The rewards in DGAP are learned from demonstrations and sampled offline data, while in RLHF are learned from human preference data that can be more expensive to collect. (ii) The reward function in DGAP is trained by a simple regression objective, while RLHF requires BT-model for explicit reward learning or DPO-style optimization for implicit reward learning (Lambert et al., 2024). (iii) We consider LLM itself as an optimizer via interactive prompting, while RLHF requires explicit optimization via RL (Ahmadian et al., 2024) or DPO (Rafailov et al., 2024; Bai et al., 2025) by updating the parameters of LLMs. Thus, our method is desirable for API-based strong LLM models, while RLHF often requires an open-sourced LLM model for reward learning and RL optimization.

## 4 RELATED WORK

**Embodied Planning with LLMs** LLMs have exhibited notable reasoning abilities in solving various tasks through in-context examples and prompting techniques like chain-of-thought (Wei et al., 2022; Vemprala et al., 2023) and tree-of-thought (Yao et al., 2023a; Zhou et al., 2023a; Feng et al., 2023). However, for embodied tasks, the in-context learning often fails as the embodied knowledge is lacking or even conflicts with that in LLMs. The existing methods introduce closed-loop feedback such as self-reflection mechanisms for self-evaluation and re-plan based on failure analysis (Madaan et al., 2023; Huang et al., 2022a; Yao et al., 2023b; Chen et al., 2024a), and external feedback for reflection (Shinn et al., 2023; Mandi et al., 2023; Zhou et al., 2023b), making them considerably costly and inefficient in querying or interactions. In contrast, we learn score function from demonstrations, eliminating the reliance on self-evaluation and external feedback. Several methods have recently considered LLMs as RL agents that can interact with the environment to collect transitions with external rewards, and perform parameter tuning by RL algorithms (Yao et al., 2024; Zhai et al., 2024). In contrast, our method is designed for API-based LLMs without parameter tuning. Similar to use, SwiftSage (Lin et al., 2023) adopts demonstrations in LLM planning while relying on imitation learning to generate actions. In contrast, we train a score function from demonstrations with augmented data, which leads to a new solution to combine the benefits of LLMs and demonstrations.

**LLM for Decision Making.** Beyond planning, LLMs can also play other important roles in embodied decision-making. (i) LLMs as a reward designer (Ma et al., 2023; Xie et al., 2023; Yu et al., 2023). The code-writing ability of LLMs can be used to generate reward codes according to the robot and task scripts. The reward function is used to train an RL policy, and the feedback from the environment can be used to perform evolutionary optimization for code generation. (ii) LLMs as a world model (Zhao et al., 2023b; Hao et al., 2023; Murthy et al., 2023). The world model is important for predicting future states and simulating long-term outcomes of actions. LLM can serve as a world model that benefits model-based RL and LLM planning. (iii) LLMs a foundation policy. LLMs can serve as a policy for imitation learning (Li et al., 2024; Brohan et al., 2023), and the policy is fine-tuned with embodied data from real-world tasks. (iv) LLM as codes generator. LLM can directly generate robot code for execution (Liang et al., 2023; Mu et al., 2024) or generate value maps to combine with model-predictive control methods (Huang et al., 2023). (v) Environment generator. LLMs can generate environments and tasks in a simulator via a closed-loop process (Wang et al., 2023a;b), which can subsequently generate training data for policy learning.

| Task Type | *Len | TDT | SFT | Reflexion | S-GPT4 | D-GPT4 | S-Llama3 | D-Llama3 |
|---|---|---|---|---|---|---|---|---|
| 1-1(L) | 107.70 | 0.71 | 15.00 | 4.22 | 97.04 | **100.00** | 40.33 | **82.67** |
| 1-2(L) | 78.60 | 0.44 | 24.40 | 10.61 | 87.04 | **92.75** | 79.00 | **91.50** |
| 1-3(L) | 88.90 | 3.88 | 32.20 | 7.78 | 72.78 | **74.00** | 59.33 | **82.67** |
| 1-4(L) | 75.20 | 0.55 | 57.45 | 0.92 | 100.00 | **100.00** | 84.00 | **90.66** |
| 2-1(M) | 21.40 | 6.16 | 9.45 | 5.92 | 99.17 | **100.00** | 76.00 | **78.67** |
| 2-2(M) | 35.20 | 6.43 | 6.75 | 28.59 | **88.17** | 80.17 | **58.00** | 46.67 |
| 2-3(L) | 65.00 | 19.87 | 5.75 | 22.37 | **95.73** | 88.33 | 76.00 | **76.00** |
| 3-1(S) | 13.60 | 40.55 | 70.00 | **100.00** | 88.67 | **91.50** | 76.00 | **78.67** |
| 3-2(M) | 20.80 | 14.26 | 48.33 | 17.45 | 55.33 | **58.00** | 100.00 | **100.00** |
| 3-3(M) | 25.60 | 10.16 | 59.50 | 72.54 | 71.90 | **78.57** | 100.00 | **100.00** |
| 3-4(M) | 29.00 | 21.65 | 69.00 | 70.22 | 77.86 | **88.14** | 100.00 | **100.00** |
| 4-1(S) | 14.60 | 41.93 | **100.00** | 64.93 | 100.00 | **100.00** | 100.00 | **100.00** |
| 4-2(S) | 8.80 | 55.76 | **100.00** | 87.27 | 100.00 | **100.00** | 100.00 | **100.00** |
| 4-3(S) | 12.60 | 27.82 | 94.45 | 16.42 | 91.67 | **100.00** | 72.33 | **76.29** |
| 4-4(S) | 14.60 | 47.15 | **100.00** | **100.00** | 100.00 | **100.00** | 100.00 | **100.00** |
| 5-1(L) | 69.50 | 6.89 | 13.45 | 7.33 | **74.59** | 73.14 | 58.00 | **78.00** |
| 5-2(L) | 79.60 | 11.86 | 44.67 | 13.00 | **93.93** | 90.57 | 35.67 | **57.33** |
| 6-1(M) | 33.60 | 15.10 | 26.25 | **70.35** | 49.40 | **57.40** | 100.00 | 78.67 |
| 6-2(S) | 15.10 | 15.70 | 53.33 | 70.67 | 100.00 | **100.00** | 100.00 | **100.00** |
| 6-3(M) | 23.00 | 5.25 | 8.00 | 15.77 | 91.48 | **92.43** | **84.67** | 68.00 |
| 7-1(S) | 7.00 | 30.00 | 11.19 | **100.00** | 95.00 | **100.00** | 100.00 | 85.71 |
| 7-2(S) | 7.00 | 8.43 | 83.33 | 67.50 | 85.00 | **85.71** | 84.67 | **92.00** |
| 7-3(S) | 8.00 | 8.34 | **100.00** | 50.00 | **93.33** | 92.71 | 80.00 | **100.00** |
| 8-1(M) | 40.00 | 3.86 | 77.87 | 2.58 | 89.00 | **100.00** | 52.00 | **100.00** |
| 8-2(S) | 16.30 | 8.00 | 33.00 | 8.00 | **68.50** | 38.50 | **61.67** | 45.00 |
| 9-1(L) | 97.00 | 2.53 | 8.00 | 50.63 | 75.00 | **75.00** | 50.00 | **57.14** |
| 9-2(L) | 84.90 | 14.66 | 73.33 | **100.00** | 70.00 | **83.33** | 66.67 | **100.00** |
| 9-3(L) | 123.10 | 9.12 | **73.33** | 70.62 | 60.00 | **71.43** | 77.67 | **88.67** |
| 10-1(L) | 130.10 | 1.51 | 53.33 | 50.90 | **92.30** | 87.71 | 43.00 | **53.00** |
| 10-2(L) | 132.10 | 1.29 | 17.00 | 23.69 | 77.60 | **78.00** | 78.00 | **84.00** |
| Short | 11.76 | 28.37 | 78.68 | 71.47 | **92.22** | 90.84 | 87.47 | **87.77** |
| Medium | 28.58 | 10.36 | 32.90 | 35.43 | 77.79 | **81.84** | 83.83 | **84.01** |
| Long | 94.30 | 6.11 | 32.55 | 30.17 | 83.00 | **84.52** | 62.31 | **78.47** |
| Overall | 49.26 | 14.66 | 49.22 | 45.34 | 84.68 | **85.91** | 76.43 | **82.96** |

Table 1: TASK PERFORMANCE ACROSS BASELINES IN SCIENCEWORLD.

## 5 EXPERIMENTS

To evaluate the effectiveness of DGAP and other baseline methods in complex embodied reasoning tasks, we employ the ScienceWorld (Wang et al., 2022) and VirtualHome (Puig et al., 2018) benchmark. They both encompass a collection of open scenarios and diverse objects for manipulation to accomplish embodied tasks.

### 5.1 SCIENCEWORLD

**Experimental Setup**  We conducted our evaluation on ScienceWorld, a virtual textual environment designed for complex science tasks that are structured with 30 different types of science experiments across 10 topics, featuring diverse locations like an art studio, kitchen, and outdoor area. Over 200 types of interactive objects and 25 action templates are included.

Our evaluation employed a series of test variations that presented unique combinations of objects and scenarios. For example, while our expert data included experiments such as freeze water, the evaluation extended to scenarios requiring the freeze mercury.

**Compared Methods**  We benchmark DGAP methodology against three kinds of approaches: **(i) Behavior Cloning-Only**: The Text Decision Transformer (TDT) leverages behavior cloning and incorporates reward-to-go as an input, which enables the model to predict actions designed to maximize future expected rewards (Chen et al., 2021). **(ii) Planning via Self-Reflection**: Techniques such as Reflexion (Shinn et al., 2023) integrate a self-reasoning mechanism within the planning pro-

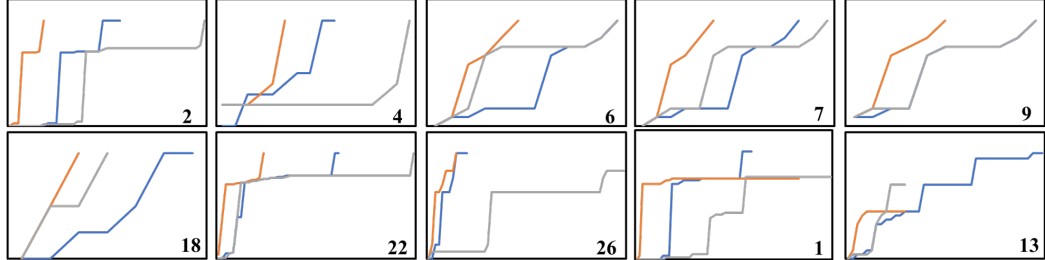

Figure 5: Visualizing trajectories of DGAP, SWIFTSAGE and ORACLE, *X*: time steps, *Y*: scores. Task identifiers are positioned at the bottom right of each figure, whose detailed information can be found in Fig. 10.

cess to enhance reasoning capabilities. **(iii) Demonstration Method**: STF is based on imitation learning on expert data. SwiftSage (Lin et al., 2023) amalgamates rapid thinking with demonstrations as our method and methodical reasoning, establishing itself as the state-of-the-art baseline in ScienceWorld, making it our primary focus of compare.

For our implementations, We used around around half number of demonstrations as ToT, SFT, and SwiftSage, with 10 to 30 trajectories per task. For Reflexion, we provided the three most relevant trajectories in the context each time, ensuring coverage of expert trajectories across tasks. We utilize Llama3-70B and GPT-4 as the foundational Large Language Models. Specifically, **S-GPT4** represents the SwiftSage method utilizing GPT-4, while **D-GPT4** denotes the DGAP strategy integrated with GPT-4. Similarly, **S-Llama3** corresponds to the SwiftSage approach adapted Llama3-70B, and **D-Llama3** signifies the DGAP method deployed with Llama3-70B. TDT and Reflexion utilize GPT-4.

**Results**  Our findings are outlined in Tab. 5.1, which elucidates the performance across thirty distinct task types. The details of tasks can be referred in Appendix. 3. Analysis of the results yields several observations: **(i)** DGAP outperforms SwiftSage in most tasks, suggesting that the external suggestion of alignment with expert data is more effective than limited internal environmental feedback. **(ii)** In tasks classified as **short**, our method shows no substantial superiority over SwiftSage. We suspect this comes from the limited pairs of short task sequences being less effective compared to the plentiful pairs in long trajectories, while short tasks primarily rely on detailed textual information to elicit LLMs' reasoning prowess. **(iii)** In addition to the success rate, DGAP demonstrates a notable enhancement in efficiency, achieving higher scores in fewer steps as validated by external expert assessments. This increased efficiency is visually represented in Fig. 5 and in Appendix 10. **(iv)** The inferential prowess of various LLMs exhibits discrepancies, with each excelling in distinct areas. For example, GPT-4 shows outstanding performance in tasks 1-1, 1-3, and 1-4, which involve changing the state of objects. Conversely, in more complex scenarios such as tasks 3-3, 3-4, and 3-5 that are related to circuits, Llama3 surpasses GPT-4, underscoring the diverse strengths of different models.

## 5.2 VIRTUALHOME

**Experimental Setup**  VirtualHome is an interactive platform to simulate complex household activities via programs and train agents to perform complete them. It also includes a Knowledge Base, Providing instructions for a diverse combination of activities, such as *put one pancake in stove and switch on stove,put two milk on kitchentable*.

We perform experimental evaluations on three distinct settings as delineated in (Li et al., 2022): **In-Distribution**, **NovelTasks**, and **NovelScenes**. They differ markedly in task complexity, which we assess using the number of action steps required for each task. Specifically, the average action steps required for **In-Distribution** are 13.4, for **NovelTasks** 25.61, and for **NovelScenes** 27.11. To ensure robustness and reliability of our findings, we replicate experiments on 60 tasks from each setting three times, thereby gathering comprehensive results. We employ **EXEC.** and **SR.** to evaluate the feasibility of the generated plans , where **EXEC.** measures whether the generated plan can be executed in the given environment, and **SR.** measures the fulfillment of task-specific goal conditions.

**Compared Methods**  We evaluate DGAP against three kinds of approaches: **(i) Planning based**: LLMs employ a method that directly generates planning results through in-context learning, such as

| | In-Distribution | | NovelScenes | | NovelTasks | |
|---|---|---|---|---|---|---|
| | EXEC. | SR. | EXEC. | SR. | EXEC. | SR. |
| **ProgPrompt** | $87.33 \pm 2.02$ | $82.33 \pm 1.76$ | $38.67 \pm 1.45$ | $32.33 \pm 1.45$ | $49.67 \pm 3.18$ | $49.00 \pm 3.21$ |
| **Inner Monologue** | $79.67 \pm 3.38$ | $79.33 \pm 3.18$ | $54.33 \pm 1.76$ | $53.33 \pm 1.76$ | $47.33 \pm 1.67$ | $46.00 \pm 1.15$ |
| **Tree Planner** | $-$ | $-$ | $\mathbf{89.33 \pm 0.17}$ | $41.67 \pm 3.20$ | $\mathbf{90.33 \pm 0.32}$ | $52.33 \pm 2.03$ |
| **DGAP-Llama3** | $90.67 \pm 0.86$ | $84.33 \pm 2.12$ | $63.00 \pm 1.68$ | $56.33 \pm 1.12$ | $78.33 \pm 1.03$ | $68.00 \pm 2.03$ |
| **DGAP-GPT4** | $\mathbf{93.33 \pm 1.76}$ | $\mathbf{88.00 \pm 2.45}$ | $71.67 \pm 1.15$ | $\mathbf{62.67 \pm 1.33}$ | $73.67 \pm 1.15$ | $\mathbf{72.17 \pm 3.18}$ |
| **DGAP-InternVL2-8B** | $84.33 \pm 1.15$ | $68.67 \pm 2.30$ | $57.06 \pm 2.11$ | $45.33 \pm 1.20$ | $52.25 \pm 1.00$ | $41.17 \pm 2.80$ |

Table 2: OVERALL PERFORMANCE DGAP AND BASELINES ACROSS VIRTUALHOME

ProgPrompt (Singh et al., 2023). **(ii) Reasoning based**: In this approach, LLMs are rendered with a specialized prompting mechanism to enhance their inference when tackling complex tasks, such as Inner Monologue (Huang et al., 2022b). **(iii) Search based**: This approach reframes the inference process into plan sampling and tree construction, thereby establishing a comprehensive and efficient pathway for task execution (Hu et al., 2023). **(iii) VLM experiments**: We have supplemented our experiments with Vision Language Models (VLMs) in the VirtualHome benchmark. Specifically, we utilized InternVL2-8B to generate the key object states within scenes, replacing the information that was previously obtained directly from the environment graph. Additionally, due to the limitations of the VLMs' field of vision, which only allows for the retrieval of objects within the current scene, we adjusted the action step length for each query. Instead of generating a full sequence for a subgoal at once, the generation now stops when the next action is [walk] (indicating a need to move to a different location) and then begin another in the new scene.

**Results** The "In-Distribution" task category is ill-suited to the Tree Planner (Hu et al., 2023) mechanism, designed to adapt to novel scene and task combinations. So, the experiments with Tree Planner in the In-Distribution setting are ignored. As shown in Tab. 8, the primary results emphasize several specific insights: **(i)** Across nearly all evaluated metrics and settings, DGAP consistently outperforms competing methodologies. This underscores the superior guidance our approach offers over environmental feedback, thereby significantly boosting success rates. **(ii)** In the majority of settings, DGAP demonstrates a minimal standard deviation. This indicates that our framework augments the embodied capabilities of LLMs and substantially improves their stability and robustness. **(iii)** Though the Tree Planner exhibits superior performance in EXEC. Due to the action tree being grounded and optimized, our method maintains a lead in success rates. This advantage stems from our continuous action evaluation loop, which refines the strategic foresight of LLMs by leveraging step-wise action-score pairs and real-time contextual inputs. **(iv)** The results indicate a relatively mild impact of using VLMs on action executability (EXEC) and a more significant influence on task success rate (SR). This suggests EXEC may rely more heavily on reasoning models rather than perception models, while SR appears to be more sensitive to the accuracy of environmental information. Additionally, we observe that among the three task categories, In-Distribution tasks are less affected by VLM-generated information, while the other two categories experience a greater impact. This suggests that familiar tasks exhibit a certain level of robustness under varying types of environmental descriptions.

Empirical evidence from experiments on two benchmarks corroborates that the DGAP method not only refines the performance of LLMs but also reduces the necessary steps and queries, thereby enhancing the efficiency and effectiveness of the embodied task's planning process.

## 6  CONCLUSION

This paper presents the Discriminator-Guided Action Optimization (DGAP) framework, addressing the challenges of complex embodied tasks which demand extensive, executable planning in dynamic scenerios. By utilizing a few demonstrations, the DGAP framework establishes a discriminator with a scoring function as real-time feedback, guiding LLMs to closely align with expert actions. Experimental results demonstrate that DGAP outperforms other baseline methods in benchmarks such as ScienceWorld and VirtualHome, showcasing superior performance and higher efficiency. The limitations of this work lie in its suboptimal performance on short-sequence tasks and the additional effort required to prepare the data. In the future, We will also investigate lightweight frameworks and adopt a generalized approach to extend their applicability to a broader range of LLM tasks.

## REPRODUCIBILITY

To ensure reproducibility, the code for our experiments is available at `https://github.com/HauffQian/DGAP`. Detailed information on models, data processing steps and experiments can be found there.

## ACKNOWLEDGEMENTS

This work is supported by National Key Research and Development Program of China (Grant No.2024YFE0210900), National Natural Science Foundation of China(Grant No. U21A20488) and National Natural Science Foundation of China (Grant No.62306242).

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

# A    THEORETICAL PROOF

## A.1    PROOF OF LEMMA 3.1

*Proof.* We recall the critic-regularized problem as follows.

$$\mathcal{L}^{\mathrm{dgap}} = \mathbb{E}_{s \sim d^{\pi}_{\mathrm{llm}}, a \sim \pi^{\mathrm{llm}}}\big[R_{\phi}(s,a)\big] - \beta D_{\mathrm{KL}}\big[\pi_{\theta}(a|s)\|\pi^{\mathrm{llm}}(a|s)\big]. \quad \mathrm{s.t.} \int_a \pi_{\theta}(a|s)da = 1. \quad (7)$$

In a constrained optimization problem, $\beta$ can be considered as a Lagrange multiplier that controls the KL-divergence between the learned policy and the basic LLM policy. Then we convert it into a Lagrangian form by introducing a factor $\alpha_s$ as

$$\mathcal{L}(\pi_{\theta}, \beta, \alpha) = \int_s d^{\pi}_{\mathrm{llm}}(s) \int_a \pi^{\mathrm{llm}}(a|s) R_{\phi}(s,a) \, dads - \beta \int_s d^{\pi}_{\mathrm{llm}}(s) D_{\mathrm{KL}}\big[\pi_{\theta}(\cdot|s)\|\pi^{\mathrm{llm}}(\cdot|s)\big] ds$$
$$+ \int_s \alpha_s \left(1 - \int_a \pi(a|s)da\right) ds, \tag{8}$$

with $\beta$ and $\alpha = \{\alpha_s | \forall s \in \mathcal{S}\}$ corresponding to the Lagrange multipliers. Differentiating $\mathcal{L}(\pi_{\theta}, \beta, \alpha)$ with respect to $\pi(a|s)$ results in

$$\frac{\partial \mathcal{L}}{\partial \pi(a|s)} = d^{\pi}_{\mathrm{llm}}(s) R_{\phi}(s,a) - \beta d^{\pi}_{\mathrm{llm}}(s) \log \pi_{\theta}(a|s) + \beta d^{\pi}_{\mathrm{llm}}(s) \log \pi^{\mathrm{llm}}(a|s) - \beta d^{\pi}_{\mathrm{llm}}(s) - \alpha_s. \quad (9)$$

Setting to zero and solving for $\pi_{\theta}(a|s)$ gives

$$\log \pi_{\theta} = \frac{1}{\beta} R_{\phi}(s,a) + \log \pi^{\mathrm{llm}}(a|s) - 1 - \frac{\alpha_s}{\beta \cdot d^{\pi}_{\mathrm{llm}}}. \tag{10}$$

Then we have

$$\pi_{\theta}(a|s) = \pi^{\mathrm{llm}}(a|s) \exp\left(\frac{1}{\beta} R_{\phi}(s,a)\right) \exp\left(-\frac{\alpha_s}{\beta} \frac{1}{d^{\pi}_{\mathrm{llm}}} - 1\right). \tag{11}$$

Since $\int_a \pi(a|s) = 1$, the second exponential term is the partition function $Z(s)$ that normalizes the conditional action distribution, as

$$Z(s) = \exp\left(\frac{\alpha_s}{\beta} \frac{1}{d^{\pi}_{\mathrm{llm}}} + 1\right) = \int_{a'} \pi^{\mathrm{llm}}(a'|s) \exp\left(\frac{1}{\beta} R_{\phi}(s,a')\right) da'. \tag{12}$$

Then the optimal policy is given by

$$\pi^{\star}_{\theta}(a|s) = \frac{1}{Z(s)} \pi^{\mathrm{llm}}(a|s) \exp\left(\frac{1}{\beta} R_{\phi}(s,a)\right), \tag{13}$$

which concludes our proof. $\qquad\square$

## A.2    PROOF OF COROLLARY 3.2

*Proof.* We remark that the objective function $\mathcal{L}^{\mathrm{dgap}}$ can also be formulated as a constrained optimization problem. Considering a tabular case with finite state and actions, we have

$$\pi^{\star}_{\theta} = \arg\max_{\pi_{\theta}} \mathbb{E}_{s \sim d^{\pi}_{\mathrm{llm}}}\left[\sum_a \pi^{\mathrm{llm}}(s,a) Q^{\pi^{\mathrm{llm}}}(s,a)\right] \quad \mathrm{s.t.} \ D_{\mathrm{KL}}\big[\pi_{\theta}(\cdot|s)\|\pi^{\mathrm{llm}}(\cdot|s)\big] \le \epsilon, \forall s, \quad (14)$$

where we use $Q^{\pi}(s,a)$ to denote $\mathbb{E}_{\pi}[R_{\phi}(s,a)]$. It is easy to check that equation 14 has the same solution as equation 7 by relaxing the hard KL constraint into a soft constraint with a coefficient $\beta$. From equation 14, we have

$$\sum_a \pi^{\star}_{\theta}(s,a) Q^{\pi^{\mathrm{llm}}}(s,a) \ge \sum_a \pi^{\mathrm{llm}}(s,a) Q^{\pi^{\mathrm{llm}}}(s,a), \forall s. \tag{15}$$

Then we have

$$
\begin{aligned}
Q^{\pi^{\text{llm}}}(s,a) &= \mathbb{E}\left[r(s_t,a_t) + \sum \pi^{\text{llm}}(a_{t+1}|s_{t+1})Q^{\pi^{\text{llm}}}(s_{t+1},a_{t+1})\Big|s_t=s,a_t=a\right] \\
&\leq \mathbb{E}\left[r(s_t,a_t) + \sum \pi_\theta^\star(a_{t+1}|s_{t+1})Q^{\pi^{\text{llm}}}(s_{t+1},a_{t+1})\Big|s_t=s,a_t=a\right] \\
&= \mathbb{E}\left[r(s_t,a_t) + \sum \pi_\theta^\star(a_{t+1}|s_{t+1})\left[r(s_{t+1},a_{t+1}) + \sum \pi^{\text{llm}}(a_{t+2}|s_{t+2})Q^{\pi^{\text{llm}}}(s_{t+2},a_{t+2})\right]\Big|...\right] \\
&\leq \mathbb{E}\left[r(s_t,a_t) + \sum \pi_\theta^\star(a_{t+1}|s_{t+1})\left[r(s_{t+1},a_{t+1}) + \sum \pi_\theta^\star(a_{t+2}|s_{t+2})Q^{\pi^{\text{llm}}}(s_{t+2},a_{t+2})\right]\Big|...\right].
\end{aligned}
\tag{16}
$$

Repeat this process to expand the equation, we have

$$
Q^{\pi^{\text{llm}}}(s,a) \leq \mathbb{E}_{\pi_\theta^\star}\left[\sum_{k=0}^{T-t-1}\gamma^k r(s_{t+k},s_{t+k})\Big|s_t=s,a_t=a\right] = Q^{\pi_\theta^\star}(s,a),
\tag{17}
$$

which concludes our proof. $\qquad\square$

# B  LM FINE-TUNING

We selected FLAN-T5-large as the base model due to its fine-tuning on datasets phrased as instructions, which improves its ability to follow and respond to instructions. According to the research detailed in (Chung et al., 2024), the model architecture includes:

- Number of layers: 24
- Number of attention heads: 16
- Hidden layer size: 1024
- Feedforward Network Size: 4096
- Activation Function: ReLU (Rectified Linear Unit)
- Total parameters: 770 million

Further we conduct imitation learning on FLAN-T5-Large using expert data in format of $(l, h_t) \to a_{\pi^{\text{oracle}(l,s_t)}}$. The data primarily encompasses three components: 1.a description of the task, 2.a record of historical actions, and 3.the actions that ought to be executed.The specifics are delineated below.

> **ScienceWorld**:
> {"input": *"Your task is to **measure the melting point of lead**, which is located around the kitchen. 10. look around.",* "Action": "go to hallway"}
> {"input": *"Your task is to **measure the melting point of lead**, which is located around the kitchen. 10. look around. 9. go to hallway.",* "Action": "open door to kitchen"}
> {"input": "Your task is to **measure the melting point of lead**, which is located around the kitchen. 10. look around. 9. go to hallway. 8. open door to kitchen.", "Action": "go to kitchen"}

> **VirtualHome**:
> {"input": *"Task: **Pay bills**. [Walk] <home_office> (1)",* "Action": "[Walk] <filing_cabinet> (1)"}
> {"input": *"Task: **Pay bills**. [Walk] <home_office> (1), [Walk] <filing_cabinet> (1)",* "Action": "[Find] <bills> (1)"}
> {"input": *"Task: **Pay bills**. [Walk] <home_office> (1), [Walk] <filing_cabinet> (1), [Find] <bills> (1)",* "Action": "[Grab] <bills> (1)"}

Then we refined FLAN-T5-large with the dataset as previously outlined, including 15k samples from ScienceWorld and 34k from VirtualHome. For the training, we employed the Adam optimizer with an epsilon value of 1e-06, a learning rate of 1e-4, and a batch size of 32. We conducted 3 training epochs comprising 25000 steps in total. We employ four A100 GPUs for conducting this task, consuming eight hours.

## C    DISCRIMINATOR TRAINING

**Data Preparation**    For training the discriminator, we collected a diverse dataset consisting of positive, negative and augmented samples, as illustrated in 3.1, Our motivation to assess the generalization necessitated the judicious use of expert data, thus preventing its overuse, particularly for tasks like *Measuring Boiling Point, Testing Conductivity, and Growing*, each initially comprising approximately 10k samples. Thus in ScienceWorld, to mitigate the dependency on expert information, we reduced the expert samples for each task to 1.5k, resulting in a total expert dataset of 45k. In parallel, we sampled 45k negative random samples. Finally, we generated 400k augmented samples, thereby creating a comprehensive pool of 500k samples. In VirtualHome, we utilized the entirety of available expert samples, given the infrequency of task repetition and a total count of only 34k. We constructed a dataset of 400k instances in a manner similar to that employed for ScienceWorld. The specific format of the data is $(l, h_t, a_t) \rightarrow q$, represented as follows:

**ScienceWorld-Expert**:
{"input": *"Your task is to **measure the melting point of lead**, which is located around the kitchen. 10. look around. Action: go to hallway"*, "Score": "**10**" }
{"input": *"Your task is to **measure the melting point of lead**, which is located around the kitchen. 10. look around. 9. go to hallway. Action: open door to kitchen"*, "Score": "**10**" }
**ScienceWorld-Random**:
{"input": *"Your task is to **measure the melting point of lead**, which is located around the kitchen. 10. look around. Action: look at art studio"*, "Score": "**0**" }
{"input": *"Your task is to **measure the melting point of lead**, which is located around the kitchen. 10. look around. 9. go to hallway. Action: put down orange"*, "Score": "**0**" }
**ScienceWorld-Augmented**:
{"input": *"Your task is to **measure the melting point of lead**, which is located around the kitchen. 10. look around. Action: look at hallway"*, "Score": "**9.03**" }
{"input": *"Your task is to **measure the melting point of lead**, which is located around the kitchen. 10. look around. 9. look at hallway. Action: open door to outside"*, "Score": "**6.13**" }
{"input": *"Your task is to **measure the melting point of lead**, which is located around the kitchen. 10. look around. 9. look at hallway. 8. open door to outside. Action: teleport to kitchen"*, "Score": "**8.87**" }
....

**VirtualHome-Expert**:
{"input": *"Task: **Pay bills**. [Walk] <home_office> (1). Action: [Walk] <filing_cabinet> (1)"*,"Score": "**10**"}
{"input": *"Task: **Pay bills**. [Walk] <home_office> (1), [Walk] <filing_cabinet> (1). Action: [Find] <bills> (1)"*, "Score": "**10**"}
**VirtualHome-Random**:
{"input": *"Task: **Pay bills**. [Walk] <home_office> (1). Action: [open] <microwave> (1)"*,"Score": "**0**"}
{"input": *"Task: **Pay bills**. [Walk] <home_office> (1), [Walk] <filing_cabinet> (1). Action: [grab] <chicken> (1)"*, "Score": "**0**"}
**VirtualHome-Augmented**:
{"input": *"Task: **Pay bills**. [Walk] <home_office> (1). Action: [walk] <kitchencabinet> (1)"*,"Score": "**8.32**"}
{"input": *"Task: **Pay bills**. [Walk] <home_office> (1), [walk] <kitchencabinet> (1). Action: [grab] <bills> (1)"*, "Score": "**7.42**"}
{"input": *"Task: **Pay bills**. [Walk] <home_office> (1), [walk] <kitchencabinet> (1), [grab] <bills> (1). Action: [walk] <livingroom>(1)"*, "Score": "**6.99**"}
.....

$$\mathcal{Q} = \begin{cases} 10 & \text{if data in } \mathcal{B}_e \\ 0 & \text{if data in } \mathcal{B}_r \\ Sim\left(a_{\hat{\pi}^{\mathrm{bc}}}, a_{\pi^{\mathrm{oracle}}}\right) * 10 & \text{if data in } \mathcal{B}_a \end{cases} \tag{18}$$

**Model Architecture**    We employ RoBERTa complemented by a linear head, which facilitates direct output for regression tasks. This configuration leverages RoBERTa's robust contextual embedding capabilities while the linear head(768,1) simplifies the mapping from embedded space to target labels, the model architecture is listed:

- Number of Layers: 12 layers

- Hidden Size: 768

- Number of Attention Heads: 12

- Feedforward Network Size: 3072

- Activation Function: GELU

- Total parameters: 125 million

**Training Procedure**    We assembled datasets containing 500k instances for ScienceWorld and 400k instances for VirtualHome seperately, each formatted as (instruction, action, score). The model was initialized with RoBERTa parameters and optimized using the AdamW optimizer a learning rate of 1e-5, a warmup rate of 0.1, and a batch size of 32. For detailed training specifications, please refer to (Tan et al., 2023). During training, the inputs comprising instructions and actions are given into the RoBERTa model to obtain the last hidden state. This state is then processed through a linear head to compute scores, which are compared against labels to determine the L2 loss. Subsequently, this loss is used to update the model parameters. We employ four A100 GPUs for conducting this task, consuming around forty hours.

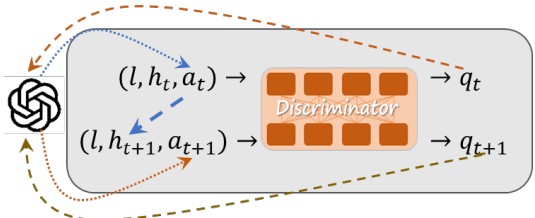

Figure 6: Illustration of applying discriminator to LLM

**Discriminator Application**    Fig. 6 depicts the functioning of a discriminator within a LLM interacting framework. Initially, the discriminator receives input tuples composed of a task decription $l$, history actions $h_t$, and response action of LLM $a_t$. It evaluates these inputs to generate a score $q_t$, which assesses the relevance to the expert response of the input tuple.

As the interaction progresses, the discriminator's inputs are advanced to the subsequent state, encapsulated by the tuple $(l, h_{t+1}, a_{t+1})$. Here, $h_{t+1}$ incorporates the previously generated action $a_t$ and based on the new context the LLM subsequently derives the new action $a_{t+1}$. In response, the discriminator calculates a subsequent score, $q_{t+1}$, for this new input.

The blue and orange dashed lines in the diagram represent information transmission at different time steps, highlighting the iterative and conditional role of the discriminator in evaluating successive actions in the sequence.

Actions generated by the Large Language Model (LLM) are not executed immediately but instead stored in an action buffer. These actions are subsequently scored by a discriminator prior to execution. If the score is below 4, a replanning process called Score-based Search is triggered, wherein the discriminator evaluates and selects the highest-scoring action from the set of valid actions for execution, as shown in Fig. 7. Conversely, if the score exceeds 8, the action is highlighted in the subsequent interaction round by including it as *'Noted: history action-score pair'* in the prompt, ensuring its prominence. This procedure ensures that the discriminator not only accesses LLM inferences but also grounds actions when necessary to prevent anomalies.

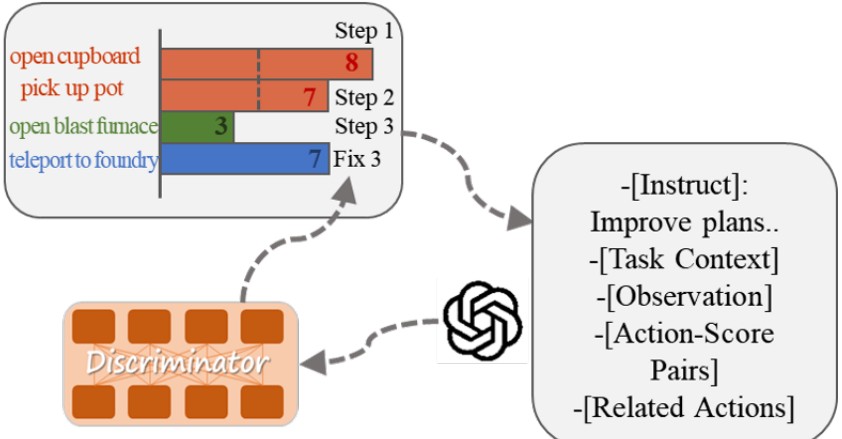

Figure 7: Illustration of score-based search

# D    STATISTICS AND DETAILS IN PLANNING

## D.1    EXPERIMENT STATICS

Table 3 provides an overview of the 30 distinct tasks within the ScienceWorld benchmark, categorized by task type, topic, and associated attributes. For the evaluation phase, we optimized time efficiency by selecting only the first 10 variations for tasks that originally had more than 10 test variations, resulting in a total of 270 task variations. This approach ensured fair and cost-effective comparisons across agents. Table 3 also includes trajectory statistics, where *Len represents the average length of the oracle agent's trajectories, offering insight into task complexity across different task types.

## D.2    BASELINES

**ScienceWorld Baselines**    We benchmark DGAP methodology against three kinds of approaches: (**i**) **Behavior Cloning-Only**: The Text Decision Transformer (TDT) leverages behavior cloning and incorporates reward-to-go as an input, which enables the model to predict actions designed to maximize future expected rewards (Chen et al., 2021). (**ii**) **Planning via Self-Reflection**: Techniques such as ReAct (Yao et al., 2023b) and Reflexion (Shinn et al., 2023) integrate a self-reasoning mechanism within the planning process to enhance reasoning capabilities. (**iii**) **Demonstration-Driven Method**: SwiftSage (Lin et al., 2023) amalgamates rapid thinking with demonstrations as our method and methodical reasoning, establishing itself as the state-of-the-art baseline in Science-World, making it our primary focus of compare. For our implementations, we utilize Llama3-70B and GPT-4 as the foundational Large Language Models. Specifically, **S-GPT4** represents the Swift-Sage method utilizing GPT-4, while **D-GPT4** denotes the DGAP strategy integrated with GPT-4. Similarly, **S-Llama3** corresponds to the SwiftSage approach adapted Llama3-70B, and **D-Llama3** signifies the DGAP method deployed with Llama3-70B. TDT ReAct and Reflexion utilize GPT-4.

**VirtualHome Baselines**    We evaluate DGAP against three kinds of approaches: (**i**) **Planning based**: LLMs employ a method that directly generates planning results through in-context learning, such as ProgPrompt (Singh et al., 2023). (**ii**) **Reasoning based**: In this approach, LLMs are rendered with a specialized prompting mechanism to enhance their inference when tackling complex tasks, such as Inner Monologue (Huang et al., 2022b). (**iii**) **Search based**: This approach reframes the inference process into plan sampling and tree construction, thereby establishing a comprehensive and efficient pathway for task execution (Hu et al., 2023).

You are giving executable advice and instructions in science experiment....

**Example task:** Your task is to use chemistry to create green paint...

- (in hallway) Action: teleport to art studio(**Score:10**) - (in art studio) Action: pour cup containing blue paint in art studio into jug(**Score:10**)
- (in art studio) Action: pour cup containing yellow paint in art studio into jug(**Score:10**)
......

In a new science experiment..
There are a few room....

**Action History:**

- 1. You move to the art studio.(**Score:10**)
- 2. In art studio, pour red paint in cup containing red paint into cup containing nothing in table(**Score:5**)
- 3. In art studio, mix paint in wood cup containing red paint(**Score:5**)
- 4. In art studio, wait(**Score:5**)
- 5. In art studio, wait(**Score:5**)

This room is called the kitchen. In it, you see...
  - a chair. On the chair is: nothing.
  - a counter. On the counter is: a bowl ···
  - a cupboard. The cupboard door is closed.
  ...

**Task Goal:**
1. You have to maximize your score and complete the task efficiently and correctly by high score action.
2. Please identify the intrinsic relationship between the action-score pairs and decide the progress corresponding to the successful experiment
3. Please review the task description and the previous observations and then answer the following questions ...
4. Please review description and observation then generate subgoal

**Question:**
1. List required materials...
2. Identify five critical subgoals focusing on specified items.
3. Indicate achieved subgoals and outline immediate next steps.
......

let's do science experiments with a sequence of actions....
In this environment, there are a few locations: ...

**Example task:** Your task is to use chemistry to create green paint...

- (in hallway) Action: TELEPORT to art studio(**Score:10**) --> You move to the art studio.
- (in art studio) Action: POUR cup containing blue paint in art studio into jug(**Score:10**) --> You pour the contents of the wood cup into the jug.
- (in art studio) Action: POUR cup containing yellow paint in art studio into jug(**Score:10**) --> You pour the contents of the wood cup into the jug.

In a new science experiment, my task is to ....

My previous 10 actions, scores and observations are as follows:

- (in hallway) You move to the art studio.(**Score:10**)
- (in art studio) In art studio, pour red paint in cup containing red paint into cup containing nothing in table(**Score:5**)
- (in art studio) In art studio, mix paint in wood cup containing red paint(**Score:5**)
- (in art studio) In art studio, wait(**Score:5**)
- (in art studio) In art studio, wait(**Score:5**)

I need to **obtain next steps with scores as high as possible.**

**Response** to Questions in Stage 1.

This room is called the kitchen. In it, you see...

Please consider the environment and action information provided above to obtain the next actions to **achieve higher scores.:**

**The ONLY allowed action types are:**
- WAIT() : wait for something to be done, for example, an object on stove to be boiled
- TELEPORT(room) : directly go to a room such as TELEPORT(kitchen)
...

Please think carefully and be concise use the below format to organize the response.
Action 1: [...] -->
 Action 2: [...] -->
 ...

Figure 8: Prompt Template in ScienceWorld

| Task Num | Type | Topic | Name | *Lens |
|---|---|---|---|---|
| 1 | 1-1 | Matter | Changes of State (Boiling) | 107.7 |
| 2 | 1-4 | Matter | Changes of State (Any) | 75.2 |
| 3 | 6-1 | Chemistry | Mixing (generic) | 33.6 |
| 4 | 6-2 | Chemistry | Mixing paints (secondary colours) | 15.1 |
| 5 | 6-3 | Chemistry | Mixing paints (tertiary colours) | 23 |
| 6 | 4-4 | Classification | Find an animal | 14.6 |
| 7 | 4-1 | Classification | Find a living thing | 14.6 |
| 8 | 4-2 | Classification | Find a non-living thing | 8.8 |
| 9 | 4-3 | Classification | Find a plant | 12.6 |
| 10 | 1-3 | Matter | Changes of State (Freezing) | 88.9 |
| 11 | 5-2 | Biology | Grow a fruit | 79.6 |
| 12 | 5-1 | Biology | Grow a plant | 69.5 |
| 13 | 8-2 | Biology | Identify life stages (animal) | 16.3 |
| 14 | 8-1 | Biology | Identify life stages (plant) | 40 |
| 15 | 9-1 | Forces | Inclined Planes (determine angle) | 97 |
| 16 | 9-2 | Forces | Friction (known surfaces) | 84.9 |
| 17 | 9-3 | Forces | Friction (unknown surfaces) | 123.1 |
| 18 | 7-1 | Biology | Identify longest-lived animal | 7 |
| 19 | 7-3 | Biology | Identify longest-lived animal | 8 |
| 20 | 7-2 | Biology | Identify shortest-lived animal | 7 |
| 21 | 2-2 | Measurement | Measuring Boiling Point (known) | 35.2 |
| 22 | 2-3 | Measurement | Measuring Boiling Point (unknown) | 65 |
| 23 | 1-2 | Matter | Changes of State (Melting) | 78.6 |
| 24 | 10-1 | Biology | Mendelian Genetics (known plants) | 130.1 |
| 25 | 10-2 | Biology | Mendelian Genetics (unknown plants) | 132.1 |
| 26 | 3-1 | Electricity | Create a circuit | 13.6 |
| 27 | 3-2 | Electricity | Renewable vs Non-renewable Energy | 20.8 |
| 28 | 3-3 | Electricity | Test Conductivity (known) | 25.6 |
| 29 | 3-4 | Electricity | Test Conductivity (unknown) | 29 |
| 30 | 2-1 | Measurement | Use Thermometer | 21.4 |
| Short (0 <*Len ≤ 20) | | | Total: 10 | 11.76 |
| Medium (20 <*Len ≤ 50) | | | Total: 8 | 28.58 |
| Long (*Len > 50) | | | Total: 12 | 94.30 |
| **Overall** | | | Total: 30 | 49.26 |

Table 3: The table presents detailed information on the 30 distinct tasks in the ScienceWorld benchmark, categorized by task number, type, topic, and name. *Len refers to the average length of the oracle agent's trajectories, based on which tasks are grouped into short, medium, and long categories, indicating the complexity and duration of each task. Additionally, the breakdown by task type (e.g., Matter, Biology, Chemistry) highlights the diversity of domains covered in the benchmark.

### D.3 PROMPT DETAILS

Our prompt design was inspired by the two-stage framework of SwiftSage (Lin et al., 2023), and further incorporate DGAP information at various places in the context(marked in red).

Our prompt was influenced by ProgPrompt (Singh et al., 2023) and Tree-Planner (Hu et al., 2023), and incorporates score-related information into the context(marked in red).

## E ADDITIONAL RESULTS AND ANALYSIS

**Ablation Study** In **ScienceWorld**, the tasks 1-15 are based on the DGAP-Llama3, whereas tasks 16-30 are according to DGAP-GPT4, both excluding content related to scores from the context

from actions import walk <obj>, grab <obj>, switchon <obj>, switchoff <obj>···
The total task goal: put_chicken_on kitchentable(id:123)#
The completed task goal:

Rule:
# rememeber if the key object INSIDE kitchencabinet···

The completed Action:
[walk] <livingroom> (262) (Score:8)
[walk] <pancake> (342) (Score:8)
[grab] <pancake> (342) (Score:9)
......

There are some examples:
The task goal: put one cupcake in stove and switch on stove
find('kitchencabinet(id:131)') (Score:10)
open('kitchencabinet(id:131)') (Score:10)
find('cupcake(id:334)') (Score:10)
grab('cupcake(id:334)') (Score:10)
......

Question:
#The currnet task goal: find_stove(id:150):
 #Remember obtain actions with scores as high as possible
Def task():......

Figure 9: Prompt Template in VirtualHome

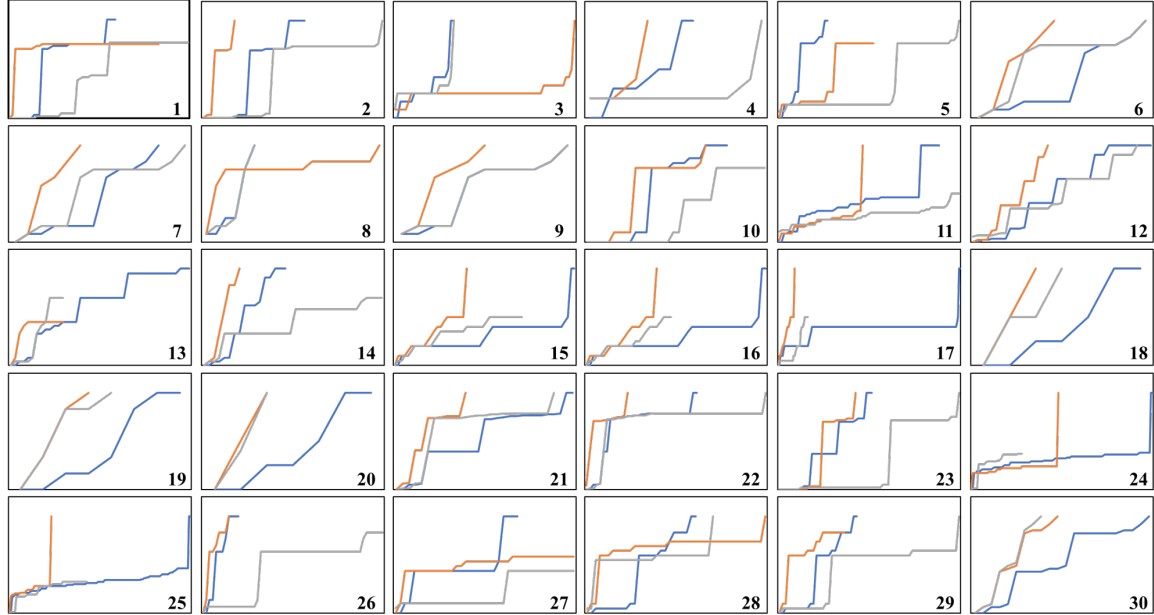

Figure 10: Visualizing trajectories of DGAP, SWIFTSAGE and ORACLE, the x-axis represents time steps, ranging from 0 to $T$, while the y-axis denotes scores, which vary from 0 to 100. Each graph illustrates the trajectories corresponding to different tasks in test variation. Task identifiers are positioned at the bottom right of each figure, whose detailed information can be found in Tab. 3

| Task | D-Llama3 | w/o Score | | Task | D-GPT4 | New Data |
|------|----------|-----------|---|------|--------|----------|
| 1 | **82.67** | 77.33 | | 16 | **83.33** | 78.57 |
| 2 | **90.66** | 76.00 | | 17 | 71.43 | **85.71** |
| 3 | **78.67** | 66.50 | | 18 | **100.00** | 100.00 |
| 4 | **100.00** | 100.00 | | 19 | **92.71** | 92.71 |
| 5 | 68.00 | **100.00** | | 20 | **85.71** | 78.57 |
| 6 | **100.00** | 85.71 | | 21 | **80.17** | 62.86 |
| 7 | **100.00** | 87.50 | | 22 | **88.33** | 50.14 |
| 8 | **100.00** | 100.00 | | 23 | **92.75** | 80.67 |
| 9 | 76.29 | **88.14** | | 24 | 87.71 | **100.00** |
| 10 | **82.67** | 65.33 | | 25 | **78.00** | 46.14 |
| 11 | **57.33** | 45.86 | | 26 | **91.50** | 83.00 |
| 12 | **78.00** | 73.43 | | 27 | **58.00** | 50.50 |
| 13 | **45.00** | 40.00 | | 28 | **78.57** | 77.80 |
| 14 | **100.00** | 100.00 | | 29 | **88.14** | 87.43 |
| 15 | 57.14 | **71.43** | | 30 | **100.00** | 61.43 |

Table 4: Ablation study of DGAP in ScienceWorld

| | In-Distribution | | NovelScenes | | NovelTasks | |
|---|---|---|---|---|---|---|
| | EXEC. | SR. | EXEC. | SR. | EXEC. | SR. |
| **ProgPrompt** | $87.33 \pm 2.02$ | $82.33 \pm 1.76$ | $38.67 \pm 1.45$ | $32.33 \pm 1.45$ | $49.67 \pm 3.18$ | $49.00 \pm 3.21$ |
| **Inner Monologue** | $79.67 \pm 3.38$ | $79.33 \pm 3.18$ | $54.33 \pm 1.76$ | $53.33 \pm 1.76$ | $47.33 \pm 1.67$ | $46.00 \pm 1.15$ |
| **Tree Planner** | – | – | $89.33 \pm 0.17$ | $41.67 \pm 3.20$ | $90.33 \pm 0.32$ | $52.33 \pm 2.03$ |
| **DGAP-Llama3** | $90.67 \pm 0.86$ | $84.33 \pm 2.12$ | $63.00 \pm 1.68$ | $56.33 \pm 1.12$ | $78.33 \pm 1.03$ | $68.00 \pm 2.03$ |
| **DGAP-GPT4** | $\mathbf{93.33 \pm 1.76}$ | $\mathbf{88.00 \pm 2.45}$ | $71.67 \pm 1.15$ | $\mathbf{62.67 \pm 1.33}$ | $73.67 \pm 1.15$ | $\mathbf{72.17 \pm 3.18}$ |
| **D-GPT4 w/o Score** | $92.33 \pm 2.19$ | $86.00 \pm 2.65$ | $70.67 \pm 2.67$ | $60.67 \pm 0.63$ | $70.33 \pm 3.06$ | $70.00 \pm 3.21$ |

Table 5: ABLATION STUDY OF DGAP AND BASELINES

and scoring thresholds mechanism for handling the exception. The results reveal that tasks 5, 9, 15, 17, and 24 exhibited better performance after the removal of score-related content, spanning both short and long task types. This phenomena likely stems from a significant deviation between the augmented data used for training the discriminator and the actual data encountered in real-world interactions in these tasks. It is noteworthy that tasks 6-3, 4-3, 9-1, 9-3, and 10-1 exhibit no significant performance improvements when evaluated against the SwiftSage baseline.

In **VirtualHome** environment, we selected DGAP-GPT4 for an ablation study due to its significantly superior performance compared to DGAP-Llama3. The results indicate that, following the removal of score-related context and the absence of scoring thresholds mechanism, both the success rate and the execution rate decreased by 1 to 2 percent. This demonstrates that DSAP is effective not only in science tasks but also in household tasks, where it provides substantial guidance.

## E.1 ALFRED EXPERIMENTS

| Methods | Seen | | Unseen | |
|---------|------|-----|--------|-----|
| | SR | GC | SR | GC |
| LLMPlanner | 0.121 | 0.267 | 0.162 | 0.402 |
| LoTaBench | 0.255 | 0.442 | 0.241 | 0.398 |
| Prompter | **0.494** | **0.560** | 0.423 | 0.537 |
| DGAP-3.5Turbo | 0.121 | 0.267 | 0.162 | 0.402 |
| DGAP-4o | 0.462 | 0.507 | **0.441** | **0.545** |
| DGAP-4o-v | 0.323 | 0.404 | 0.381 | 0.414 |
| DGAP-InternVL2-8B | 0.236 | 0.291 | 0.246 | 0.391 |

Table 6: Performance comparison across methods on ALFRED seen and unseen tasks.

**Analysis of Methods and Results**

Table 6 compares several methods on **Seen** and **Unseen** environments using **Success Rate (SR)** and **Goal Completion (GC)** metrics. **LLMPlanner** and **LoTaBench** rely only on LLMs and expert data for in-context learning. **Prompter** integrates semantic map information and combines low-level parsing with LLMs. **DGAP-3.5Turbo** uses `gpt-3.5-turbo-0125` as the backbone model. **DGAP-4o** is based on Azure's GPT-4o model. **DGAP-4o-v** shares the same GPT-4o backbone but incorporates updated environmental maps during the planning phase. **DGAP-InternVL2-8B** shares the same settings as DGAP-4o-v, including updated environmental maps.

**DGAP-4o** only uses textual environment information and one-fifth of the expert data for discriminator's training, yet achieves competitive results. In Seen environments, DGAP achieves strong performance (SR: 0.462, GC: 0.507), slightly behind Prompter. However, in Unseen environments, DGAP outperforms Prompter in GC (0.545 vs. 0.537), demonstrating better generalization. On the other hand, **DGAP-4o-v**, which incorporates RGB images into the planning process, shows a decline in performance. It's likely attributed to two main factors: (1) the prompts and discriminator have not been fully optimized to utilize visual information. (2) the RGB images may introduce ambiguous or redundant information rather than render location information. These observations suggest that the integration of visual data requires more refined strategies to fully unlock its potential benefits.

**Weaknesses of DGAP:** Despite its strengths, there are two notable limitations that affect its performance in certain scenarios:

1. **Inconsistent Ground Truth**: The ground truth action sequences occasionally lack logical coherence or completeness. For example, in the "Turn on the bedroom lamp" task, the ground truth includes extraneous steps such as "Pick up CellPhone" before toggling the lamp. In the "Put a plate in a cabinet" task, additional steps like "Place plate in the fridge on the top shelf" are required. Task "Put a bowl with a pencil in it on the desk," where the ground truth includes an unnecessary intermediate step to "Put the bowl on the Shelf." There are various such cases which introduce noise and reduce alignment between the generated actions and expected outcomes for LLM planning.

2. **Ambiguity in Object Location**: For tasks involving similar objects, selecting one randomly among them can often suffice to meet the task's objective. However, in scenarios requiring precise identification of specific storage locations (e.g., determining the correct cabinet where an object is stored), DGAP faces challenges due to the absence of explicit spatial or semantic distinctions in the textual input. This limitation impacts its ability to perform accurate object placement, leading to reduced performance on tasks that rely on spatial reasoning and clear differentiation between similar storage units. .

E.2 VH-LONG-TASK

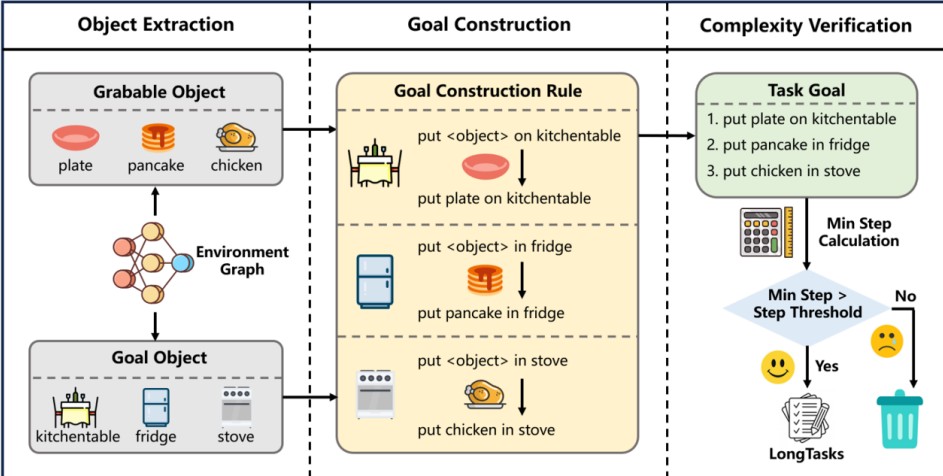

Figure 11: The framework of task constructing

**Extended Task Settings and Dataset Construction** To better reflect the high-dimensional and continuous tasks commonly encountered in real-world scenarios, which are typically multi-goal in nature, we extended the task settings (Wu et al., 2024) in the VirtualHome environment to include **self-constructed long tasks**. By concatenating multiple task objectives, we created tasks exceeding **60 steps**, providing a more complex and realistic evaluation of long-horizon planning. The statistics of these extended tasks are summarized in the table below.

| Dataset | Goals Number | Action Step | Objects Number | Objects Variety |
|---|---|---|---|---|
| In-Distribution | 3.40 | 26.58 | 5.32 | 3.57 |
| NovelScenes | 3.39 | 26.27 | 5.32 | 3.56 |
| NovelTasks | 3.99 | 27.02 | 4.97 | 3.40 |
| LongTasks | **9.74** | **77.01** | **15.79** | **8.50** |

Table 7: Dataset statistics for various tasks in VirtualHome.

| | Extreme-LONG | | NovelScenes | | NovelTasks | |
|---|---|---|---|---|---|---|
| | EXEC. | SR. | EXEC. | SR. | EXEC. | SR. |
| **ProgPrompt** | $42.95 \pm 1.22$ | $16.03 \pm 1.28$ | $38.67 \pm 1.45$ | $32.33 \pm 1.45$ | $49.67 \pm 3.18$ | $49.00 \pm 3.21$ |
| **Inner Monologue** | $48.45 \pm 1.05$ | $16.16 \pm 1.35$ | $54.33 \pm 1.76$ | $53.33 \pm 1.76$ | $47.33 \pm 1.67$ | $46.00 \pm 1.15$ |
| **Tree Planner** | $\mathbf{85.76 \pm 1.40}$ | $19.07 \pm 1.79$ | $\mathbf{89.33 \pm 0.17}$ | $41.67 \pm 3.20$ | $\mathbf{90.33 \pm 0.32}$ | $52.33 \pm 2.03$ |
| **DGAP-GPT4o** | $78.23 \pm 1.15$ | $\mathbf{67.25 \pm 2.10}$ | $71.67 \pm 1.15$ | $\mathbf{62.67 \pm 1.33}$ | $73.67 \pm 1.15$ | $\mathbf{72.17 \pm 3.18}$ |

Table 8: OVERALL PERFORMANCE DGAP AND BASELINES ACROSS VH-LONG-TASK

Experimental results show that our method significantly outperforms the baseline in success rate, highlighting its superiority in handling high-complexity, long-horizon tasks and demonstrating the generalization capability of the discriminator.

## E.3 DISCRIMINATOR EVALUATION

| Methods | Out-of-Distribution | | In-Distribution | | PER | OPer |
|---|---|---|---|---|---|---|
| | Acc | Var | Acc | Var | | |
| 4x | 9.17 | 0.11 | 9.28 | 0.08 | 85.91 | - |
| 2x | 9.16 | 0.11 | 9.28 | 0.08 | 85.91 | - |
| 0.4x | 8.77 | 0.15 | 9.19 | 0.12 | 83.25 | - |
| basis | 8.08 | 0.36 | 8.16 | 0.36 | 82.26 | 83.53 |
| basiswithforce | 8.06 | 0.38 | 8.15 | 0.26 | 82.31 | 84.51 |
| basiswithforcebio | 8.91 | 0.20 | 9.01 | 0.17 | 84.98 | 86.68 |
| 1x(ours) | 9.08 | 0.15 | 9.21 | 0.11 | 85.91 | - |

Table 9: Performance comparison of discriminator across data with different volumn and categories, with including PER and OPer metrics.

## E.4 SEARCH-BASED METHODS EVALUATION

We utilized the Vanna framework to construct the RAG pipeline, incorporating the **bge-large** model as the embedding backbone. The content matching process was based on a combination of task instructions and the initial environmental observations. We subsequently present its recall performance and evaluate its effectiveness within the context of SwiftSage's deliberative reasoning, a highly effective in-context learning framework for ScienceWorld. The results are shown in Table 10

## E.5 QUALITATIVE ANALYSIS

**Feedback Mechanism** S-GPT4 failed in **Task8** due to its strict adherence to the rule prohibiting focus on counterparts when conditions are not fulfilled, exposing the limitations of relying solely

| Task Type | *Len | S-GPT4 | D-GPT4 | Recall (%) | Performance |
|-----------|------|--------|--------|-----------|-------------|
| 1-1(L) | 107.70 | 97.04 | **100.00** | 91.8 | 90.89 |
| 1-2(L) | 78.60 | 87.04 | **92.75** | 89.2 | 89.20 |
| 1-3(L) | 88.90 | 72.78 | **74.00** | 87.2 | 57.20 |
| 1-4(L) | 75.20 | 100.00 | **100.00** | 90.6 | 89.41 |
| 2-1(M) | 21.40 | 99.17 | **100.00** | 94.8 | 93.99 |
| 2-2(M) | 35.20 | **88.17** | 80.17 | 88.9 | 82.90 |
| 2-3(L) | 65.00 | **95.73** | 88.33 | 86.4 | 86.40 |
| 3-1(S) | 13.60 | 88.67 | 91.50 | 97.8 | **100.00** |
| 3-2(M) | 20.80 | 55.33 | **58.00** | 89.6 | 49.60 |
| 3-3(M) | 25.60 | 71.90 | **78.57** | 90.0 | 68.44 |
| 3-4(M) | 29.00 | 77.86 | 88.14 | 91.5 | **92.66** |
| 4-1(S) | 14.60 | 100.00 | **100.00** | 99.4 | **100.00** |
| 4-2(S) | 8.80 | 100.00 | **100.00** | 98.2 | **100.00** |
| 4-3(S) | 12.60 | 91.67 | **100.00** | 96.2 | **100.00** |
| 4-4(S) | 14.60 | 100.00 | **100.00** | 98.5 | **100.00** |
| 5-1(L) | 69.50 | **74.59** | 73.14 | 85.2 | 55.20 |
| 5-2(L) | 79.60 | **93.93** | 90.57 | 90.8 | 61.58 |
| 6-1(M) | 33.60 | 49.40 | 57.40 | 88.5 | **58.50** |
| 6-2(S) | 15.10 | 100.00 | **100.00** | 98.5 | **100.00** |
| 6-3(M) | 23.00 | 91.48 | **92.43** | 91.4 | 90.78 |
| 7-1(S) | 7.00 | 95.00 | **100.00** | 97.1 | **100.00** |
| 7-2(S) | 7.00 | 85.00 | 85.71 | 92.4 | **93.63** |
| 7-3(S) | 8.00 | **93.33** | 92.71 | 95.8 | **97.96** |
| 8-1(M) | 40.00 | 89.00 | **100.00** | 94.4 | 96.28 |
| 8-2(S) | 16.30 | **68.50** | 38.50 | 93.3 | **91.76** |
| 9-1(L) | 97.00 | 75.00 | **75.00** | 85.9 | 35.90 |
| 9-2(L) | 84.90 | 70.00 | **83.33** | 86.1 | 69.10 |
| 9-3(L) | 123.10 | 60.00 | **71.43** | 85.8 | 61.80 |
| 10-1(L) | 130.10 | **92.30** | 87.71 | 92.4 | 82.74 |
| 10-2(L) | 132.10 | 77.60 | **78.00** | 85.9 | 72.90 |
| Short | 11.76 | **92.22** | 90.84 | 97.3 | 98.12 |
| Medium | 28.58 | 77.79 | **81.84** | 90.5 | 79.14 |
| Long | 94.30 | 83.00 | **84.52** | 88.3 | 71.03 |
| Overall | 49.26 | 84.68 | **85.91** | 90.2 | 82.29 |

Table 10: Task performance comparison for S-GPT4, D-GPT4 with RAG

on the LLM, though the action is reasonable to some extent. In contrast, DGAP-GPT4 succeeded by leveraging discriminator feedback and re-planning through the LLM, which selected alternative yet equally reasonable actions to adaptively execute a comprehensive sequence. This included identifying, transporting, and arranging multiple items to fulfill all task requirements, emphasizing the importance of integrating robust feedback mechanisms and adaptive planning for complex, multi-step tasks, as shown in Figure 12.

**Enhanced Attention** The discriminator's use of additional score labels enhances attention on relevant examples and prior actions, improving environmental interaction understanding and directly boosting planning quality without relying on re-planning. This is evident in **Task25**. As shown in Figure 13, where DGAP-generated actions are efficiently aligned with task objectives, focusing on the electric buzzer early and following a logical path to complete the circuit with minimal redundancy. In contrast, the other sequence exhibits inefficiencies, including unnecessary connections and delays, highlighting the limitations of planning without adequate score attention. This comparison underscores the discriminator's role in enabling precise and efficient task execution.

**Your task is to find a(n) plant. First, focus on the thing. Then, move it to the yellow box in the bedroom**

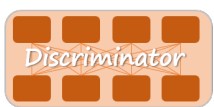

DGAP-GPT4

SwiftSage-GPT4

1. look around
2. teleport to greenhouse
3. focus on flower pot 1
**Failure**

-1.You move to the greenhouse.(Score:7)
-2. focus on flower pot 1 (Score: 5) →**replan!**
- 2. pick up flower pot 1(Score:9) --> You move the flower pot 1 to the inventory.
- 3. You move to the bedroom.(Score:7)
- 4. In bedroom, move flower pot to yellow box(Score:7) --> You move the flower pot 1 to the yellow box.
- 5. In greenhouse, pick up flower pot 6(Score:7) --> You move the flower pot 6 to the inventory.
- 6. Repeat the above 1 times.
- 7. In bedroom, move flower pot 6 to yellow box(Score:5)→**replan!**
- 7. Repeat the above 1 times.
……….
- 18. You move to the bedroom.(Score:8)
- 19. In bedroom, move flower pot 8 to yellow box(Score:8) --> You move the flower pot 8 to the yellow box

Figure 12: The visualization of the task8

**Your task is to turn on the electric buzzer. First, focus on the electric buzzer. Then, create an electrical circuit that powers it on.**

SwiftSage-GPT4

1. look around
2. teleport to workshop
3. focus on electric buzzer, which is necessary to proceed with task of turning it on
4. focus on electric buzzer
5. connect anode of battery to terminal 1 of black wire
6. connect battery anode to black wire terminal 1
7. connect terminal 2 of black wire to anode of electric buzzer
8. connect black wire terminal 2 to anode in electric buzzer
9. connect battery cathode to red wire terminal 1
10. connect terminal 2 of red wire to cathode of electric buzzer
11. connect red wire terminal 2 to cathode in electric buzzer
12. Wait
… **Failure, no circuit(color mismatch)**

DGAP-GPT4

1. look around
2. teleport to workshop
3. focus on electric buzzer to ensure all subsequent actions are directed towards activating it
4. focus on electric buzzer
5. connect battery anode to black wire terminal 1
6. connect battery cathode to orange wire terminal 1
7. connect battery cathode to terminal 1 in yellow wire
8. connect black wire terminal 2 to cathode in electric buzzer
9. connect orange wire terminal 2 to anode in electric buzzer
10. connect terminal 2 in yellow wire to anode in buzzer

**Success**

Figure 13: The visualization of the task25

