# OpenReview forum: "Discriminator-Guided Embodied Planning for LLM Agent"
_ICLR.cc/2025/Conference — ICLR 2025 Poster_

### Official Review · Reviewer_f2Sa · 2024-10-23

**Soundness:** 3
**Presentation:** 2
**Contribution:** 2
**Rating:** 6
**Confidence:** 3

**Summary:**

This paper proposes an LLM action plan optimization framework based on discriminators to enhance the policy performance in long-term tasks with high generalization ability, where a small number of demonstrations is required to guide the optimization process. This framework first learns a score function via the discriminator with limited set of demonstrations, where the LLM is prompt to generate actions to maximize the score for optimization. The experimental results demonstrate the effectiveness of the proposed method.

**Strengths:**

+ The topic of LLM-based embodied planning is of great interests in embodied AI especially for general-purpose robots.
+ The theoretical formulation to link the proposed method and critic-regularized optimization in RL brought some insights on embodied planning optimization
+ The performance shows the method outperforms the baselines by a large margin.

**Weaknesses:**

- As introduced in the Introduction Section, the proposed framework aims to bring the long-term reasoning ability from LLMs with task grounding without harming the generalization ability. The discriminator is trained on limited collected data compared with the pre-trained LLMs, and I am not sure whether the discriminator can be generalized well. More proofs are required.

- The groundtruth score for offline data in data collection is evaluated by a sentence embedding model. The quality of the sentence embedding model might significantly influence the performance of the groundtruth, where I doubt the noise of groundtruth will affect the model performance.

- The writing needs to be improved. The proposed method contains a lot of techniques. I think the most important contribution, which I guess might be the discriminator training and usage, should be clearly emphasized.

- Some qualitative results such as the planning results should be visualized to give some more intuition of the benefits brought by the proposed method.

- More analysis to the experimental results especially the performance differences between short and long sequences should be discussed, as the generalization ability on different tasks (short and long-horizon) is both important.

**Questions:**

See Weakness.

---

> ### Author Response · Authors · 2024-11-22
>
> **Q1: The generalization of discriminator**
>
> **R1:** Thanks for your insightful suggestion. We would like to tackle this propose from two perspectives:
>
> 1. **Versatility of the Discriminator**:
>    To demonstrate this, we have conducted experiments on diverse benchmarks, including ALFRED and VH-Long-Task. The detailed results can be found in Appendix E.1 and E.2.
> - **ALFRED**
>
> | Methods               | Seen SR | Seen GC | Unseen SR | Unseen GC |
> |-----------------------|---------|---------|-----------|-----------|
> | **LLMPlanner[10]**           | 0.121   | 0.267   | 0.162     | 0.402     |
> | **LoTaBench[12]**            | 0.255   | 0.442   | 0.241     | 0.398     |
> | **Prompter[11]**             | **0.494**   | **0.560**   | 0.423     | 0.537     |
> | **DAPG-3.5Turbo**        | 0.121   | 0.267   | 0.162     | 0.402     |
> | **DAPG-4o**              | 0.462   | 0.507   | **0.441**     | **0.545**     |
> | **DAPG-4o-v**            | 0.323   | 0.404   | 0.381      | 0.414       |
> | **DAPG-InternVL2-8B**    | 0.236   | 0.291   | 0.246     | 0.391     |
>    - **VH-LONG-TASK**
>
> | Method               | Extreme-LONG EXEC. | Extreme-LONG SR. |
> |----------------------|---------------------|------------------|
> | **ProgPrompt**       | 42.95 ± 1.22       | 16.03 ± 1.28     |
> | **Inner Monologue**  | 48.45 ± 1.05       | 16.16 ± 1.35     |
> | **Tree Planner**     | **85.76 ± 1.40**   | 19.07 ± 1.79     |
> | **DGAP-GPT4o**       | 78.23 ± 1.15       | **67.25 ± 2.10** |
>
>
> 2. **Impact of Data Volume on Discriminator Performance**:
>
>  we conducted experiments to analyze the model's performance under different conditions:
>
> - **Experiments with different types of task combinations**:
>   - **Accuracy (Acc)**: Average score of expert demonstrations within the training data (GroundTruth: 10).
>   - **Accuracy varience(Acc Var)**: Varience of average score of expert demonstrations within the training data.
>   - **OOD Accuracy (OAcc)**: Average score of expert demonstrations out of the training data (GroundTruth: 10).
>   - **OOD Accuracy varience(OAcc Var)**: Varience of average score of expert demonstrations out of the training data.
>
>
>
> | Methods              | OAcc | OAcc Var | Acc  | Acc Var |
> |----------------------|-------------------------|--------------------------|---------------------|---------------------|
> | 4x                  | 9.17                    | 0.11                     | 9.28                | 0.08                |
> | 2x                  | 9.16                    | 0.11                     | 9.28                | 0.08                |
> | 0.4x                | 8.77                    | 0.15                     | 9.19                | 0.12                |
> | 1x(Ours)            | 9.08                   | 0.15                     | 9.21                | 0.11                |
>
> The experiments involve varying training data volumes (4x, 2x, 1x, and 0.5x), where 1x corresponds to 20% of expert data. It can be observed that the marginal gains from using additional samples beyond our baseline are limited, while the discriminator demonstrates strong performance on OOD data.

---

> ### Author Response · Authors · 2024-11-22
>
> **Q2: Groundtruth Noise**
>
> **R2:** Thank you for your insightful question.
> Labeling ground truth in planning tasks remains a sophisticated and evolving challenge. Approaches such as Monte Carlo search trees (MCST) are often employed to construct outcome-supervised value functions, while others utilize large language models (LLMs) for scoring or fine-tuning to derive reward models.
>
> Although the limited scope of expert data necessitates training smaller models for data augmentation, which may introduce noise, some studies [5,6] have highlighted that even noisy augmented data can contribute to improved generalization and effectiveness. This is particularly evident in reinforcement learning [7,8], where such strategies have been shown to offer significant benefits.
>
> We have conducted an additional evaluation of our discriminator's accuracy, and the results are as follows:
>
> - **Acc and OAcc in R1**: The discriminator achieved an average score of **9.21 and 9.08** (Groundtruth score: **10**).
> - **Scoring distribution of action lists**: We use discriminator to score all the actions in the list.
>   - Probability of groundtruth action ranking in the top 3: **76.53%**.
>   - Probability of groundtruth action ranking in the top 5: **91.26%**.
> - **Average score on successfully planned trajectories**: Mean score of **8.632**.
>
> Indeed, this question addresses a critical and nuanced point. And We are currently researching heterogeneous methods constructed from a subset of expert and preference data to mitigate these biases. We hope this can address your concern.
>
> **Q3: The writing**
>
> **R3:** Thanks for your suggestion. We have revised the manuscript to improve the writing and clearly emphasize the discriminator training and usage as the key contribution.

---

> > ### Author Response · Authors · 2024-11-22
> >
> > **Q4: Qualitative Results**
> >
> > **R4:** We demonstrate the differences from three perspectives，and we have included the details in Appendix E.5 due to content limitations and hope you will take the time to review them：
> >
> > - **Feedback Mechanism for Low-Scored Actions**
> >   - In cases where the discriminator assigns a low score to an otherwise optimal action, the system resorts to the LLM for re-planning, selecting an **alternative yet equally reasonable** action.
> >
> > - **Enhanced Attention through Score Labels**
> >   - The discriminator leverages additional score labels to increase the attention weight on examples and past actions. This mechanism improves the understanding of environmental interactions and, in certain cases, directly enhances the planning quality without relying on re-planning.
> >
> > - **Improved Efficiency via Action Filtering**
> >   - By employing the discriminator for action filtering, unnecessary "wait" actions are removed during data cleaning, leading to a notable improvement in overall task efficiency as shown in Appendix Figure.10.

---

> > > ### Author Response · Authors · 2024-11-22
> > >
> > > **Q5: Analysis of generalization on different tasks**
> > >
> > > **R5:** Thank you for your insightful question. Based on the results presented in our manuscript, our approach performs well on most short tasks but struggles significantly with task 8-2, achieving only 38.5 points. This underperformance impacts the overall evaluation, making it less competitive than the SwiftSage method. We also supplemented additional experiments as stated in **R1**. To clarify the performance gap in task 8-2, we provide a detailed example and explanation below.
> > >
> > > | **Step** | **Butterfly-Failed**                                      | **Butterfly-Done**                                      | **Bear-Failed**                                  | **Bear-Done**                                  |
> > > |----------|--------------------------------------------------------------------------------|--------------------------------------------------------------------------------|------------------------------------------------------------------------------|------------------------------------------------------------------------------|
> > > | 1        | look around                                                                    | look around                                                                    | look around                                                                  | look around                                                                  |
> > > | 2        | teleport to outside                                                           | teleport to outside                                                           | teleport to outside                                                         | teleport to outside                                                         |
> > > | 3        | focus on butterfly eggs (failed action, no matching input)                    | focus on butterfly egg                                                        | focus on brown bear cub (failed action, no matching input)                  | focus on baby brown bear                                                    |
> > > | 4        | focus on butterfly egg                                                        | wait (simulate hatching time)                                                 | focus on baby baby brown bear                                               | wait (simulate growth time)                                                 |
> > > | 5        | wait                                                                           | focus on caterpillar                                                          | wait                                                                         | focus on juvenile brown bear                                                |
> > > | 6        | repeat steps 3 and 4 (multiple failures focusing on larvae stage)             | wait (simulate growth time)                                                   | focus on juvenile brown bear                                                | wait (simulate juvenile growth time)                                        |
> > > | 7        | focus on caterpillar                                                          | focus on butterfly pupa                                                       | wait                                                                         | focus on adult brown bear                                                   |
> > > | 8        | wait                                                                           | wait (simulate transformation to adult butterfly)                             | wait                                                                         |                                                                              |
> > > | 9        | focus on butterfly pupa                                                                         | focus on adult butterfly                                                      | focus on adult adult brown bear                                             |
> > >
> > > We hope this example illustrates that while our method closely aligns with intended actions, it faced challenges due to strict time constraints in life science experiments, where "wait" actions are tightly regulated. The additional "wait" actions, coupled with the discriminator's limitations in distinguishing them effectively, contributed to the failure. Notably, handling "wait" actions constitutes a critical aspect of training data construction. By extensively refining "wait" data, DGAP achieves significantly fewer average steps per task compared to other methods, even outperforming the ground truth in this metric. This failure underscores the need for more precise embodied data construction and more effort.

---

### Official Review · Reviewer_ESa8 · 2024-11-03

**Soundness:** 4
**Presentation:** 3
**Contribution:** 4
**Rating:** 8
**Confidence:** 4

**Summary:**

The paper proposes a framework named Discriminator-Guided Action Optimization (DGAP), which combines the long-term reasoning capabilities of large language models (LLMs) with task-specific grounding under guidance. The authors introduce a simple discriminator learned with a limited number of demonstrations and. validate the effectiveness of this approach through experiments. The paper also provides a detailed theoretical explanation of the relationship between this method and critic-regularized optimization in RL.

**Strengths:**

1. The paper employs a score function to quantitatively assess the planning effectiveness of the LLM, making it more reasonable compared to previous works.
2. The paper provides a detailed theoretical derivation to model the problem and establish its relationship with critic-regularized optimization.
3. The paper compares various types of planners, including reasoning-based and search-based approaches.

**Weaknesses:**

1. The experiments in the paper use the ScienceWorld and VirtualHouse simulators. However, it lacks the inclusion of widely used simulators for embodied planning tasks, such as ALFRED (https://arxiv.org/abs/1912.01734).
2. The proposed method does not achieve optimal performance on short-sequence tasks; its advantages are primarily evident in long-sequence tasks.

**Questions:**

1. Could the authors consider experimenting with datasets like ALFRED? Due to its widespread use, ALFRED includes both LLM-based and RL-based methods, providing a valuable basis for further comparison and analysis.
2. According to my understanding, the authors implicitly store task-specific knowledge in the discriminator through training. It would be beneficial to compare this approach with explicitly stored, search-based methods to better demonstrate the discriminator's advantages.

---

> ### Author Response · Authors · 2024-11-22
>
> **Q1: Alfred**
> **R1:** Thank you for your insightful feedback. The discriminator serves as a modular component during inference. For handling visual and physical information, a feasible approach is either leveraging a VLM or fine-tuning a diffusion model such as RDT[2] or GR-Series[3,4]. Also, prior research has demonstrated the strong alignment between visual and linguistic modalities in semantics[5,6]. This suggests that extending language-based settings to incorporate visual and tactile modalities serves as a valid approach.
>
> On this purpose, we conducted experiments on the ALFRED benchmark, where we retrained the discriminator using ALFRED's text data and evaluated its performance using **GPT-3.5**, **GPT-4o**, and **Intern-VL2-8B** as base models. A comparison with several mainstream baselines is summarized as follows:
> - **Experimental setup**:
>   - **GPT-3.5 Turbo / GPT-4o**: These models acquire environment information and feedback in textual form to perform reasoning.
>   - **GPT-4o-v / Intern-VL2-8B**: These models utilize task instructions (text) and images to gather environment information and feedback for reasoning.
>
>
> | Methods               | Seen SR | Seen GC | Unseen SR | Unseen GC |
> |-----------------------|---------|---------|-----------|-----------|
> | **LLMPlanner[10]**           | 0.121   | 0.267   | 0.162     | 0.402     |
> | **LoTaBench[12]**            | 0.255   | 0.442   | 0.241     | 0.398     |
> | **Prompter[11]**             | **0.494**   | **0.560**   | 0.423     | 0.537     |
> | **DAPG-3.5Turbo**        | 0.121   | 0.267   | 0.162     | 0.402     |
> | **DAPG-4o**              | 0.462   | 0.507   | **0.441**     | **0.545**     |
> | **DAPG-4o-v**            | 0.323   | 0.404   | 0.381      | 0.414       |
> | **DAPG-InternVL2-8B**    | 0.236   | 0.291   | 0.246     | 0.391     |
>
> - **Results**:
>   Detailed information about the experiment is in Appendix E.1. **DGAP-4o** only uses textual environment information and one-fifth of the expert data for discriminator's training, yet achieves competitive results. In Seen environments, DGAP achieves strong performance (SR: 0.462, GC: 0.507), slightly behind Prompter. However, in Unseen environments, DGAP outperforms Prompter in GC (0.545 vs. 0.537), demonstrating better generalization. On the other hand, **DGAP-4o-v**, which incorporates RGB images into the planning process, shows a decline in performance. It's likely attributed to two main factors: (1) the prompts and discriminator have not been fully optimized to utilize visual information. (2) the RGB images may introduce ambiguous or redundant information rather than render location information. These observations suggest that the integration of visual data requires more refined strategies to fully unlock its potential benefits.
>
>
> To fully address this concern, we are conducting additional experiments including fine-tune a vision-language model using image-action pairs as a new data-augmentor and further adjust the discriminator structure to be multimodal. Hopefully we can update it in this period. This updated discriminator is designed to **score** `<instruction, image, action>` triplets.

---

> ### Author Response · Authors · 2024-11-22
>
> **Q2：Compared with search-based methods**
>
> **R2.** Thank you for your valuable feedback. We conducted comparative analyses of the planning with explicitly stored knowledge, search-based methods. And we construct a RAG(retrival augmented) framework. We sincerely hope you to review the detailed results and further comparisons provided in **Appendix E.4** as limitation of content here. Specifically, we evaluated:
>
> - **RAG Recall Performance**:
>   - For shorter tasks, RAG demonstrates better performance with higher recall rates.
>   - When retrieved samples are closely aligned with the target task, the success rate is significantly higher.
>
> - **Performance with RAG Samples in ICL**:
>   - Shorter tasks benefit more from retrieved samples when integrated into in-context learning (ICL).
>   - Planning performance is robust due to data availability in such scenarios, especially with well-matched samples.
>
> - **Analysis of Behavior**:
>   - For short tasks, we observed that in many cases, simply imitating the retrieved accurate task trajectories is sufficient to complete the tasks. However, for long tasks, while the initial steps of imitation tend to be highly precise, the performance of search methods declines once environmental feedback causes deviations from the expected trajectory.
>   - The discriminator demonstrates advantages in diverse tasks, especially for long sequence tasks, process evaluation and feedback is effective to ensure stable performance throughout the task trajectory.
>
> We hope these findings demonstrate that while search-based methods can be effective when task-relevant data is available, particularly for short tasks, our approach excels in generalization, especially for long and complex tasks.

---

> > ### Author Response · Authors · 2024-11-22
> >
> > **Q3: About performance difference**
> >
> > **R4:** Thank you for your insightful question. Based on the results presented in our manuscript, our approach performs well on most short tasks but struggles significantly with task 8-2, achieving only 38.5 points. This underperformance impacts the overall evaluation, making it less competitive than the SwiftSage method. Notably, ICL-based methods are inherently advantageous for short tasks, as the attention mechanisms allow them to effectively capture critical information within a limited scope. To clarify the performance gap in task 8-2, we provide a detailed example and explanation below.
> >
> > | **Step** | **Butterfly-Failed**                                      | **Butterfly-Done**                                      | **Bear-Failed**                                  | **Bear-Done**                                  |
> > |----------|--------------------------------------------------------------------------------|--------------------------------------------------------------------------------|------------------------------------------------------------------------------|------------------------------------------------------------------------------|
> > | 1        | look around                                                                    | look around                                                                    | look around                                                                  | look around                                                                  |
> > | 2        | teleport to outside                                                           | teleport to outside                                                           | teleport to outside                                                         | teleport to outside                                                         |
> > | 3        | focus on butterfly eggs (failed action, no matching input)                    | focus on butterfly egg                                                        | focus on brown bear cub (failed action, no matching input)                  | focus on baby brown bear                                                    |
> > | 4        | focus on butterfly egg                                                        | wait (simulate hatching time)                                                 | focus on baby baby brown bear                                               | wait (simulate growth time)                                                 |
> > | 5        | wait                                                                           | focus on caterpillar                                                          | wait                                                                         | focus on juvenile brown bear                                                |
> > | 6        | repeat steps 3 and 4 (multiple failures focusing on larvae stage)             | wait (simulate growth time)                                                   | focus on juvenile brown bear                                                | wait (simulate juvenile growth time)                                        |
> > | 7        | focus on caterpillar                                                          | focus on butterfly pupa                                                       | wait                                                                         | focus on adult brown bear                                                   |
> > | 8        | wait                                                                           | wait (simulate transformation to adult butterfly)                             | wait                                                                         |                                                                              |
> > | 9        | focus on butterfly pupa                                                                         | focus on adult butterfly                                                      | focus on adult adult brown bear                                             |                                                                              |
> >
> > We hope this example illustrates that while our method closely aligns with intended actions, it faced challenges due to strict time constraints in life science experiments, where "wait" actions are tightly regulated. The additional "wait" actions, coupled with the discriminator's limitations in distinguishing them effectively, contributed to the failure. Notably, handling "wait" actions constitutes a critical aspect of training data construction. By extensively refining "wait" data, DGAP achieves significantly fewer average steps per task compared to other methods, even outperforming the ground truth in this metric. This failure underscores the need for more precise embodied data construction and more effort.

---

### Official Review · Reviewer_SJiW · 2024-11-03

**Soundness:** 3
**Presentation:** 3
**Contribution:** 3
**Rating:** 6
**Confidence:** 3

**Summary:**

The paper introduces a novel framework named Discriminator-Guided Action Optimization (DGAP), which aims to improve the performance of Large Language Models (LLMs) in embodied planning tasks. By leveraging a small set of demonstrations to train a discriminator, DGAP facilitates the optimization of LLM-generated action plans through step-wise signals, leading to better alignment with optimal actions. The proposed method is tested on challenging benchmarks like ScienceWorld and VirtualHome, demonstrating superior performance and efficiency compared to existing methods.

**Strengths:**

1. The integration of a discriminator to guide LLMs in generating higher-quality action plans is a creative and promising approach.
2. The paper establishes a connection between DGAP and critic-regularized optimization in reinforcement learning, providing a solid theoretical foundation for the method.
3. Extensive experiments are conducted on well-known benchmarks, showcasing the practical benefits of DGAP over prior methods.
4. The paper is well-written, with clear explanations and logical organization, making it easy to follow the technical details and experimental results.

**Weaknesses:**

1. While the paper demonstrates the effectiveness of DGAP in specific environments, it would be beneficial to explore its scalability to more complex and diverse scenarios. Future work could investigate how the method performs in real-world settings with higher-dimensional state spaces.
2. The reliance on a limited set of demonstrations might limit the generalizability of the model. It would be valuable to analyze how the performance changes with varying amounts of demonstration data and whether the method can adapt to unseen tasks effectively.

**Questions:**

See above.

---

> ### Author Response · Authors · 2024-11-22
>
> **Q1: Scalability to more complex and diverse scenarios**
>
> **R1.** Thank you for your insightful feedback. We address this issue from two perspectives:
>
> **i. Multi-Modal Experiments**:
> We supplemented our study with multi-modal experiments conducted on the ALFRED benchmark. The discriminator was retrained using ALFRED's text data and evaluated with **GPT-3.5**, **GPT-4o**, and **Intern-VL2-8B** as base models. A comparison with several mainstream baselines is outlined below:
>
> | Methods               | Seen SR | Seen GC | Unseen SR | Unseen GC |
> |-----------------------|---------|---------|-----------|-----------|
> | **LLMPlanner[10]**           | 0.121   | 0.267   | 0.162     | 0.402     |
> | **LoTaBench[12]**            | 0.255   | 0.442   | 0.241     | 0.398     |
> | **Prompter[11]**             | **0.494**   | **0.560**   | 0.423     | 0.537     |
> | **DAPG-3.5Turbo**        | 0.121   | 0.267   | 0.162     | 0.402     |
> | **DAPG-4o**              | 0.462   | 0.507   | **0.441**     | **0.545**     |
> | **DAPG-4o-v**            | 0.323   | 0.404   | 0.381      | 0.414       |
> | **DAPG-InternVL2-8B**    | 0.236   | 0.291   | 0.246     | 0.391     |
>
> - **Experimental setup**:
>   - **GPT-3.5 Turbo / GPT-4o**: These models process environment information and feedback in textual form to perform reasoning.
>   - **GPT-4o-v / Intern-VL2-8B**: These models combine task instructions (text) and images to gather environment information and feedback for reasoning.
>
> - **Results**:
>   Detailed information about the experiment is in Appendix E.1. **DGAP-4o** only uses textual environment information and one-fifth of the expert data for discriminator's training, yet achieves competitive results. In Seen environments, DGAP achieves strong performance (SR: 0.462, GC: 0.507), slightly behind Prompter. However, in Unseen environments, DGAP outperforms Prompter in GC (0.545 vs. 0.537), demonstrating better generalization. On the other hand, **DGAP-4o-v**, which incorporates RGB images into the planning process, shows a decline in performance. It's likely attributed to two main factors: (1) the prompts and discriminator have not been fully optimized to utilize visual information. (2) the RGB images may introduce ambiguous or redundant information rather than render location information. These observations suggest that the integration of visual data requires more refined strategies to fully unlock its potential benefits.
>
> **ii. Task Diversity**
> To better reflect high-dimensional continuous tasks encountered in **real-life scenarios**, which are typically **multi-goal**, we extended the task settings to include **self-constructed long tasks** in the VirtualHome environment. By concatenating different task objectives, we constructed tasks exceeding **60 steps**, with the task statistics summarized in the table below. The details are shown in Appendix E.2.
>
> | Dataset          | Goals Number | Action Step | Objects Number | Objects Variety |
> |------------------|--------------|-------------|----------------|-----------------|
> | In-Distribution  | 3.40         | 26.58       | 5.32           | 3.57            |
> | NovelScenes      | 3.39         | 26.27       | 5.32           | 3.56            |
> | NovelTasks       | 3.99         | 27.02       | 4.97           | 3.40            |
> | **LongTasks**    | **9.74**     | **77.01**   | **15.79**      | **8.50**        |
>
>
>
> | Method               | Extreme-LONG EXEC. | Extreme-LONG SR. |
> |----------------------|---------------------|------------------|
> | **ProgPrompt**       | 42.95 ± 1.22       | 16.03 ± 1.28     |
> | **Inner Monologue**  | 48.45 ± 1.05       | 16.16 ± 1.35     |
> | **Tree Planner**     | **85.76 ± 1.40**   | 19.07 ± 1.79     |
> | **DGAP-GPT4o**       | 78.23 ± 1.15       | **67.25 ± 2.10** |
>
>
>
> Experimental results show that our method significantly outperforms the baseline in success rate, highlighting its superiority in handling high-complexity, long-horizon tasks and demonstrating the generalization capability of the discriminator.
>
>
> We hope these two additional results demonstrate that our method exhibits adaptability across tasks, regardless of multi-modal inputs, long sequences, or diverse domains. Our method also ensures task-level executability, under which high-level plans can be reliably translated into low-level control signals via a fixed parser.

---

> > ### Author Response · Authors · 2024-11-22
> >
> > **Q2: Reliance of demonstrations**
> >
> > **R2.** Thank you for the question. To address this concern, we conducted experiments to analyze the model's performance under different conditions:
> >
> > - **Experiments with different types of task combinations**:
> >   - **Accuracy (Acc)**: Average score of expert demonstrations within the training data (GroundTruth: 10).
> >   - **Accuracy varience(Acc Var)**: Varience of average score of expert demonstrations within the training data.
> >   - **OOD Accuracy (OAcc)**: Average score of expert demonstrations out of the training data (GroundTruth: 10).
> >   - **OOD Accuracy varience(OAcc Var)**: Varience of average score of expert demonstrations out of the training data.
> >   - **Planning Performance (PER)**: Task score for tasks within the training set.
> >   - **OOD Planning Performance (OPER)**: Task score for tasks out of the training set.
> >
> > | Methods              | OAcc | OAcc Var | Acc | Acc Var | PER   | OPer   |
> > |----------------------|-------------------------|--------------------------|---------------------|---------------------|-------|--------|
> > | 4x                  | 9.17                    | 0.11                     | 9.28                | 0.08                | 85.91 | -      |
> > | 2x                  | 9.16                    | 0.11                     | 9.28                | 0.08                | 85.91 | -      |
> > | 0.4x                | 8.77                    | 0.15                     | 9.19                | 0.12                | 83.25 | -      |
> > | basis               | 8.08                    | 0.36                     | 8.16                | 0.36                | 82.23 | 83.53  |
> > | basiswithforce      | 8.06                   | 0.38                    | 8.15               | 0.36               | 82.31 | 84.51  |
> > | basiswithforcebio   | 8.91                   | 0.20                    | 9.01               | 0.17               | 84.94 | 86.68  |
> > | 1x(ours)            | 9.08                   | 0.15                    | 9.21               | 0.11               | 85.91 | -      |
> >
> >
> > The experiments involve varying training data volumes (4x, 2x, 1x, and 0.5x), where 1x corresponds to 20% of expert-labeled data. The dataset is divided into three key subsets: basis tasks (tasks 1, 3, 6) with 11,433 samples, focusing on foundational capabilities; biological tasks (tasks 4, 5, 7, 8, 10) with 20,432 samples, contributing the largest and most diverse subset; and force tasks with 9,661 samples, targeting physical reasoning.
> >
> > It can be observed that the marginal gains from using additional samples beyond our baseline are limited, while the discriminator demonstrates strong performance on OOD data. Furthermore, **diversity in task types** proves critical for the generalization ability of the discriminator. A lack of diversity in the training data leads to a notable decline in discriminator accuracy and significantly impacts overall planning performance. Notably, OOD scores naturally exceed overall scores as its partil categories.

---

### Official Review · Reviewer_Em5T · 2024-11-06

**Soundness:** 3
**Presentation:** 4
**Contribution:** 2
**Rating:** 6
**Confidence:** 5

**Summary:**

The authors proposed an embodied agent framework to optimize LLM action plans at each step. They achieve so by employing a trained discriminator that learns a score function of alignment between LLM predicted actions and the optimal action, the author. They demonstrated the effectiveness of this policy by conducting experiments on benchmarks in ScienceWorld and VirtualHome, and outperformed previous methods.

**Strengths:**

-The proposed framework is effective that it achieves superior results in the benchmarks studied.

-The experiments and ablation studies in this study is thorough and the baselines compared to are extensive.

-The presentation of this paper is good, the paragraphs are well written and easy to follow

**Weaknesses:**

-The novelty of this proposed framework is unclear to me. What is the fundamental differences of the motivation and the framework between this work and previous language grounding work (e.g. Saycan)?

-The generalizability of the proposed method. If I understand correctly, the method designed does not involve fine-tuning a LLM rather it use a discriminator to capture the domain knowledge of the embodied agent. How do you see this discriminator generalize to more complicated scenarios as the granularity of visual/physical environment
is far beyond than the text modality can capture.

-The choice of benchmark. I respect your choice of experiment benchmarks. However, there are many embodied agent planning frameworks on benchmarks like ALFRED. If you cannot include them as baselines, please discuss the differences.

**Questions:**

See weaknesses.

---

> ### Author Response · Authors · 2024-11-22
>
> **Q1:The novelty and difference**
>
> **R1:** Thanks for the question. We would like begin by introducing the motivation, followed by three typical related works representing distinct grounding approaches, and elaborating on the differences. **DGAP** serves as a generalized framework designed to address the generalization and efficiency challenges of complex embodied tasks (>25 steps) by designing a discriminator to provide **process scores** and integrating it with **critic optimization** in planning.
>
>
> Then we discuss the differences between our approach and related works in terms of method design, data sources, and model construction:
> - **SayCan**[9]
>   - SayCan evaluates actions by integrating high-level and low-level scores.
>     - The generative decoding of the LLM produces high-level explanations, which are subsequently integrated with the RL-trained low-level affordance function to evaluate action feasibility.
>     - However, this approach demands extensive ground-truth collection and relies on delicate model designs, constraining its generalization capabilities. For example, the affordance model in SayCan is limited to only 15 object categories.
>   - In contrast, our method:
>     - Leverages the generalization capability of large models (reasoning models) and the domain-specific knowledge of small models (discriminator models).
>     - Introduces score labels to bridge these two models.
>     - Addresses generalization and efficiency issues under limited expert data.
>
> - **LLM Planner**[10]
>   - The LLM planner employs a retrieval-based approach using few-shot examples to adapt to new environments.
>     - However, the static nature of the information introduced during inference restricts the method's adaptability across different tasks.
>     - The method is constrained by the availability of expert examples.
>   - In contrast, our method:
>     - Is less sensitive to the quantity of examples (see Appendix E.3).
>     - Augments a limited set of examples to train a discriminator, dynamically providing step-level signals to the LLM.
>
> - **ReAd**[1]
>   - Read is perhaps the most closely related work, as it also provides scores during the process to represent task contributions.
>   - However, compared to ReAd:
>     - Their method uses an MLP-based structure, while we adopt a **RoBERTa with regression head** architecture, which offers stronger semantic capture capabilities, particularly for long-text scenarios.
>     - ReAd relies on interaction-based data collection, whereas our approach achieves higher sample utilization and greater data collection efficiency by fine-tuning a small model and employing beam search sampling.

---

> ### Author Response · Authors · 2024-11-22
>
> **Q2: Generalize discriminator to visual/physical environment**
>
> **R2：** Thank you for your insightful feedback. The discriminator serves as a modular component during inference. For handling visual and physical information, a feasible approach is either leveraging a VLM or fine-tuning a diffusion model such as RDT[2] or GR-Series[3,4]. Also, prior research has demonstrated the strong alignment between visual and linguistic modalities in semantics[5,6]. This suggests that extending language-based settings to incorporate visual and tactile modalities serves as a valid approach.
>
> On this purpose, we conducted experiments on the ALFRED benchmark, where we retrained the discriminator using ALFRED's text data and evaluated its performance using **GPT-3.5**, **GPT-4o**, and **Intern-VL2-8B** as base models. A comparison with several mainstream baselines is summarized as follows:
> - **Experimental setup**:
>   - **GPT-3.5 Turbo / GPT-4o**: These models acquire environment information and feedback in textual form to perform reasoning.
>   - **GPT-4o-v / Intern-VL2-8B**: These models utilize task instructions (text) and images to gather environment information and feedback for reasoning.
>
> | Methods               | Seen SR | Seen GC | Unseen SR | Unseen GC |
> |-----------------------|---------|---------|-----------|-----------|
> | **LLMPlanner[10]**           | 0.121   | 0.267   | 0.162     | 0.402     |
> | **LoTaBench[12]**            | 0.255   | 0.442   | 0.241     | 0.398     |
> | **Prompter[11]**             | **0.494**   | **0.560**   | 0.423     | 0.537     |
> | **DAPG-3.5Turbo**        | 0.121   | 0.267   | 0.162     | 0.402     |
> | **DAPG-4o**              | 0.462   | 0.507   | **0.441**     | **0.545**     |
> | **DAPG-4o-v**            | 0.323   | 0.404   | 0.381      | 0.414       |
> | **DAPG-InternVL2-8B**    | 0.236   | 0.291   | 0.246     | 0.391     |
>
> - **Results**:
>   Detailed information about the experiment is in Appendix E.1. **DGAP-4o** only uses textual environment information and one-fifth of the expert data for discriminator's training, yet achieves competitive results. In Seen environments, DGAP achieves strong performance (SR: 0.462, GC: 0.507), slightly behind Prompter. However, in Unseen environments, DGAP outperforms Prompter in GC (0.545 vs. 0.537), demonstrating better generalization. On the other hand, **DGAP-4o-v**, which incorporates RGB images into the planning process, shows a decline in performance. It's likely attributed to two main factors: (1) the prompts and discriminator have not been fully optimized to utilize visual information. (2) the RGB images may introduce ambiguous or redundant information rather than render location information. These observations suggest that the integration of visual data requires more refined strategies to fully unlock its potential benefits.
>
>
> To fully address this concern, we are conducting additional experiments including fine-tune a vision-language model using image-action pairs as a new data-augmentor and further adjust the discriminator structure to be multimodal. Hopefully we can update it in the period. This updated discriminator is designed to **score** `<instruction, image, action>` triplets.

---

> ### Author Response · Authors · 2024-11-25
>
> Dear Reviewer Em5T,
>
> As the deadline of the discussion period draws near, we would greatly appreciate your attention to our rebuttal. We would like to know whether we have adequately addressed your concerns. Your feedback is crucial to us, and we value the opportunity to address any concerns you have raised. If you have any further questions, we are more than happy to discuss.
>
> Thank you for your time and consideration.
>
> Best regards,
>
> Authors

---

> > ### Comment · Reviewer_Em5T · 2024-11-26
> >
> > Thank you for the response and additional experiments. My concerns are addressed. It looks like the model performance on ALFRED is far from SOTA but given the short time I imagine there might be rooms to improve. I would recommend you to include more experiments and analysis on this benchmark. I now recommend this paper to be weakly accepted.

---

> > > ### Author Response · Authors · 2024-11-27
> > >
> > > Thanks for the reply! We appreciate your time and efforts during the review and discussion. 😄

---

### Author Response · Authors · 2024-11-22

We sincerely thank all the reviewers for their valuable feedback and insightful comments. In response to the reviewers' suggestions, we have conducted several complementary experiments to address their concerns. The results of these additional experiments are summarized below:

1. \[Adding new **evaluation analysis** of the discriminator]
2. \[Adding new experiments in **ALFRED**]
3. \[Constrcting and evaluating **extreme long tasks** in virtualhome]
4. \[Adding a new baseline of **search-based methods** with full demonstration data]
5. \[Ongoing: A new multimodal discriminator training]

According to the result, our method demonstrates better (or at least comparable) performance to the competitors in all the above settings. We also clarify some questions the reviewers mentioned, including but not limited to:


1. \[The novelty and difference of our method]: DGAP leverages discriminator feedback and critic optimization to improve task generalization and efficiency, offering process-level scoring and adaptability with limited data, setting it apart from methods like SayCan[9], LLMPlanner[10], and ReAd[1].
2. \[Generalization to Visual/Physical Domains]: The modular design of the discriminator facilitates its extension to multimodal tasks. Initial results on ALFRED indicate promising adaptability and competitive generalization in unseen scenarios.
3. \[Performance difference between short and long tasks]: While DGAP performs consistently across tasks of varying lengths, challenges remain in addressing complex temporal constraints, such as handling "wait" actions, which require more refined data construction.
4. \[Groundtruth Noise]: Despite noise in augmented data, DGAP demonstrates resilience, effectively utilizing score signals to achieve generalization.
5. \[Qualitative Analysis]: DGAP effectively enhances planning by leveraging process-level scoring to focus on past actions and environmental interactions, while also demonstrating the ability to discern conditions to re-planning.

We deeply appreciate the anonymous reviewers for their insightful feedback and constructive suggestions, which have substantially improved the quality of our research. Also, we hope our response can be carefully considered and welcome any useful suggestions.


**Reference**

[1] Zhang, Yang, et al. "Towards efficient llm grounding for embodied multi-agent collaboration." arXiv preprint arXiv:2405.14314 (2024).

[2] Liu, Songming, et al. "RDT-1B: a Diffusion Foundation Model for Bimanual Manipulation." arXiv preprint arXiv:2410.07864 (2024).

[3] Cheang, Chi-Lam, et al. "Gr-2: A generative video-language-action model with web-scale knowledge for robot manipulation." arXiv preprint arXiv:2410.06158 (2024).

[4] Wu, Hongtao, et al. "Unleashing large-scale video generative pre-training for visual robot manipulation." arXiv preprint arXiv:2312.13139 (2023).

[5] Cubuk, Ekin D., et al. "Autoaugment: Learning augmentation strategies from data." Proceedings of the IEEE/CVF conference on computer vision and pattern recognition. 2019.

[6] Wu, Sen, et al. "On the generalization effects of linear transformations in data augmentation." International Conference on Machine Learning. PMLR, 2020.

[7] Raileanu, Roberta, et al. "Automatic data augmentation for generalization in reinforcement learning." Advances in Neural Information Processing Systems 34 (2021): 5402-5415.

[8] Pinneri, Cristina, et al. "Equivariant Data Augmentation for Generalization in Offline Reinforcement Learning." arXiv preprint arXiv:2309.07578 (2023).

[9] Brohan, Anthony, et al. "Do as i can, not as i say: Grounding language in robotic affordances." Conference on robot learning. PMLR, 2023.

[10] Song, Chan Hee, et al. "Llm-planner: Few-shot grounded planning for embodied agents with large language models." Proceedings of the IEEE/CVF International Conference on Computer Vision. 2023.

[11] Inoue, Yuki, and Hiroki Ohashi. "Prompter: Utilizing large language model prompting for a data efficient embodied instruction following." arXiv preprint arXiv:2211.03267 (2022).

[12] Choi, Jae-Woo, et al. "Lota-bench: Benchmarking language-oriented task planners for embodied agents." arXiv preprint arXiv:2402.08178 (2024).

---

### Meta-Review · Area_Chair_PnDz · 2024-12-22

**Metareview:**

This work proposes a Discriminator-Guided Action Optimization (DGAP) to enhance embodied planning. All reviewers consistently recommended accepting this work. AC agrees that this work is interesting and deserves to be published on ICLR 2O25. The reviewers did raise some valuable concerns that should be addressed in the final camera-ready version of the paper. The authors are encouraged to make the necessary changes in the final version.

**Additional Comments On Reviewer Discussion:**

All reviewers consistently recommended accepting this work.

---

### Decision · Program_Chairs · 2025-01-22

Accept (Poster)